# A Periodic Bayesian Flow for Material Generation

**Hanlin Wu**[1,2*]    **Yuxuan Song**[1,3*]    **Jingjing Gong**[1]    **Ziyao Cao**[1,3]    **Yawen Ouyang**[1]
**Jianbing Zhang**[4]    **Hao Zhou**[1]    **Wei-Ying Ma**[1]    **Jingjing Liu**[1]
[1] Institute of AI Industry Research (AIR), Tsinghua University
[2] School of Vehicle and Mobility, Tsinghua University
[3] Dept. of Comp. Sci. & Tech., Tsinghua University
[4] School of Artifcial Intelligence, Nanjing University
`wuhl24@mails.tsinghua.edu.cn`,
`{songyuxuan,zhouhao,ouyangyawen,jjliu}@air.tsinghua.edu.cn`

## Abstract

Generative modeling of crystal data distribution is an important yet challenging task due to the unique periodic physical symmetry of crystals. Diffusion-based methods have shown early promise in modeling crystal distribution. More recently, Bayesian Flow Networks were introduced to aggregate noisy latent variables, resulting in a variance-reduced parameter space that has been shown to be advantageous for modeling Euclidean data distributions with structural constraints (Song et al., 2023). Inspired by this, we seek to unlock its potential for modeling variables located in non-Euclidean manifolds *e.g.* those within crystal structures, by overcoming challenging theoretical issues. We introduce CrysBFN, a novel crystal generation method by proposing a periodic Bayesian flow, which essentially differs from the original Gaussian-based BFN by exhibiting non-monotonic entropy dynamics. To successfully realize the concept of periodic Bayesian flow, CrysBFN integrates a new entropy conditioning mechanism and empirically demonstrates its significance compared to time-conditioning. Extensive experiments over both crystal ab initio generation and crystal structure prediction tasks demonstrate the superiority of CrysBFN, which consistently achieves new state-of-the-art on all benchmarks. Surprisingly, we found that CrysBFN enjoys a significant improvement in sampling efficiency, *e.g.*, 200× speedup (10 *v.s.* 2000 steps network forwards) compared with previous diffusion-based methods on MP-20 dataset. Code is available at `https://github.com/wu-han-lin/CrysBFN`.

## 1 Introduction

Deep generative models, with their strong ability to approximate data distribution with complex geometries, have recently emerged as a promising approach to de novo drug design (Hoogeboom et al., 2022), protein engineering (Shi et al., 2022), and material science (Liu et al., 2017). To discover new functional materials (Wang et al., 2023; Peng et al., 2022), there has been an active line of research on crystal generative modeling (Ren et al., 2021; Hoffmann et al., 2019; Noh et al., 2019; Court et al., 2020; Yang et al., 2023; Nouira et al., 2018). Recent diffusion-based models learns through an iterative reverse process with multi-level noise perturbation, and has been demonstrated as a powerful tool for capturing complex geometries of crystals. Studies show that these models can generate crystal samples with realistic structures that well satisfy physical constraint (Xie et al., 2021; Jiao et al., 2023; 2024; Lin et al., 2024).

Despite promising results, significant challenges persist. The search space for crystal structures grows exponentially with the number of atoms, while thermodynamically stable materials represent only a small fraction (Miller et al., 2024). This presents challenges in the multi-step generation process, the variance of which might cause structures to deviate from stable distributions. Moreover,

---

*Equal Contribution. Correspondence to Hao Zhou (zhouhao@air.tsinghua.edu.cn) and Yawen Ouyang (ouyangyawen@air.tsinghua.edu.cn).

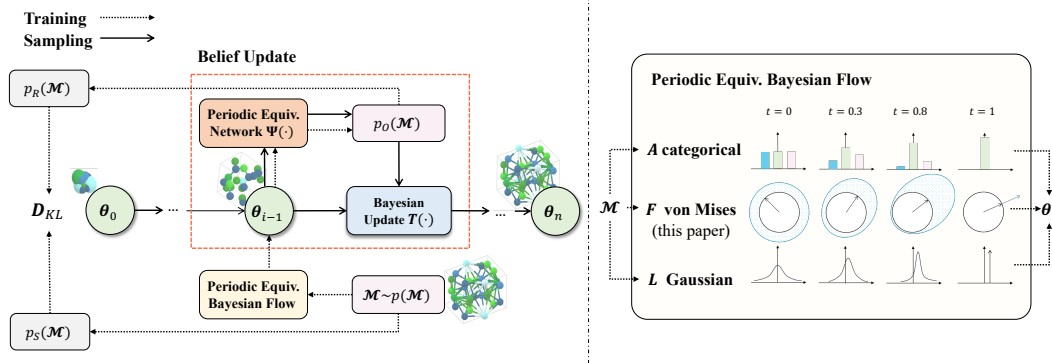

Figure 1: **Framework of CrysBFN**. Left: overview of training and sampling process. At training time, the network receives $\boldsymbol{\theta}_{i-1}$ from Bayesian flow based on data distribution, and tries to improve the belief $\boldsymbol{\theta}_{i-1}$ over the groundtruth $\mathcal{M}$ by outputting an estimated distribution $p_O$ and minimizing the gap between estimation and groundtruth. At sampling time with the trained network, the uninformative prior $\boldsymbol{\theta}_0$ is gradually improved by belief updates until $\boldsymbol{\theta}_n$ with high fidelity. Right: illustration of the periodic equivariant Bayesian flow.

the current widely-adapted diffusion-based approaches (Jiao et al., 2023; 2024) for crystal structure modeling tend to learn the score function of wrapped normal distribution for periodic variables, where the approximation of a sum of infinite terms is needed which could bring in extra bias. Recently, BFN (Graves et al., 2023) has been successfully applied to the geometry generative modeling of molecules (Song et al., 2023), a scenario that shares the above-mentioned challenges similar to crystal generation, by modeling in a much lower variance parameter space. However, the periodic geometry of crystals differs from that of small molecules and raises significant challenges.

To tackle these challenges, this paper aim to break the barrier of extending the paradigm of BFN into those variables located non-Euclidean space, *e.g.*, atom fractional coordinates in crystal structure. We introduce the first non-Euclidean Bayesian flow over the periodic space, *i.e.* the hyper-torus. To successfully implement such concept, we introduce a generalized training paradigm based on simulation of the Bayesian flow and further propose a non-auto-regressive equivalent formulation of Bayesian flow distribution that guarantees computational efficiency. By integrating all these innovations, we introduce CrysBFN, the first periodic E(3) equivariant Bayesian flow network designed for crystal generation. Extensive experiments demonstrate the significant superiority of CrysBFN over current methods in both sampling quality and efficiency.

Our contributions can be summarized as follows:

- We present the first periodic Bayesian flow in non-Euclidean space (hyper-torus) with a novel training paradigm and entropy conditioning mechanism tackling the unprecedented and pivotal non-additive accuracy theoretical challenge.

- We introduce the first periodic-E(3) equivariant Bayesian flow networks for crystal generation tasks with appealing theoretical guarantees.

- Extensive experiments demonstrate that CrysBFN consistently outperforms previous methods on both ab initio crystal generation (99.1% COV-P on Carbon-24) and crystal structure prediction tasks (64.35% match rate on MP-20). Efficiency experiments on MP-20 prove that CrysBFN enjoys a **200× sampling efficiency** with performance on par with previous Diffusion-based methods.

## 2 RELATED WORK

Modeling and generating stable materials with data-driven approaches has been applied to discovering new functional materials (Peng et al., 2022). One line of approaches indirectly models crystal space by transforming crystals into human-designed representations (Ren et al., 2021; Hoffmann et al., 2019; Noh et al., 2019; Court et al., 2020; Yang et al., 2023), though the encoding and decoding process often leads to physical geometry loss In contrast, another line of research directly models crystals in the sample space, drawing inspiration from the success of Diffusion models (Ho et al.,

2020b; Song et al., 2020a; Song & Ermon, 2019). For instance, CDVAE (Xie et al., 2021) and SyMat (Luo et al., 2024b) employ score-matching (Song & Ermon, 2019) to learn scores for generating stable materials, while their modeled distribution lacks geometric invariance (Zhang et al., 2023). DiffCSP (Jiao et al., 2023) addresses this by transforming Cartesian atom coordinates into fractional coordinates, introducing the periodic E(3) equivariance of crystals, and designing an equivariant diffusion crystal generation model based on periodic diffusion (Jing et al., 2022). More recently, Miller et al. (2024) applies Riemannian Flow Matching (Chen & Lipman, 2023) to crystal generation tasks offering improved sampling efficiency, while at the expense of quality [1]. However, we argue that these methods struggle to balance sampling quality and efficiency due to insufficient guidance during each transition from the noise prior to the data distribution. This issue is particularly pronounced for crystals, where thermodynamically stable materials constitute only a small fraction of the search space (Miller et al., 2024). For instance, early generation states $x_{t-1}$ with low confidence should be retained less than later states to achieve the next state $x_t$.

In this work, we propose to use BFN (Graves et al., 2023) to model crystals in a principally different way. BFN provides a framework to accurately update each generation state according to its entropy/confidence, the effectiveness of which has been proved in Song et al. (2023). However, there are no established explorations on the challenging topic of non-Euclidean BFN which is essential to many real-world applications Jing et al. (2022); Jumper et al. (2021). To address the above issues, in this paper, we build a non-Euclidean Bayesian flow from scratch, identifying and tackling the non-additive accuracy via introducing a novel entropy conditioning mechanism.

## 3 PRELIMINARIES

**Crystal Representation and Related Manifold** Crystals can be represented as a structure composed of infinite, periodic and repeating unit cells defined by a triplet $\mathcal{M} = (\boldsymbol{A}, \boldsymbol{F}, \boldsymbol{L})$. Denote the number of atoms in the unit cell as $N$, $\boldsymbol{A} = (\boldsymbol{a}_1, \boldsymbol{a}_2, \ldots, \boldsymbol{a}_N) \in \mathcal{S}^{K \times N}$ is the representation of atom types with the length of vocabulary $K$, and every such one-hot discrete variable locates in the simplex $\mathcal{S}^K$ represented by:

$$\mathcal{S}^K \stackrel{\text{def}}{=} \{\boldsymbol{s} \in \mathbb{R}^K | \sum_{i=1}^{K} s_i = 1, s_i \geq 0, i = 1, \ldots, K\} \tag{1}$$

which requires the designed generative path for $\boldsymbol{A}$ should be well defined on the simplex. Following Jiao et al. (2023), $\boldsymbol{F} = [\boldsymbol{f}_1, \boldsymbol{f}_2, \ldots, \boldsymbol{f}_N] \in [0, 1)^{3 \times N}$ is the fractional coordinates of atoms located in quotient space $\mathbb{R}^{3 \times N}/\mathbb{Z}^{3 \times N}$ equivalent to the hypertorus $\mathbb{T}^{3 \times N}$ (Jing et al., 2022). The hypertorus $\mathbb{T}^{3 \times N}$ can be represented as the Cartesian product of $3 \times N$ toruses $\mathbb{T}^1$:

$$\mathbb{T}^1 \stackrel{\text{def}}{=} \{\boldsymbol{z} \in \mathbb{R}^2 : ||\boldsymbol{z}|| = 1\} \tag{2}$$

And $\boldsymbol{L} = [\boldsymbol{l}_1, \boldsymbol{l}_2, \boldsymbol{l}_3] \in \mathbb{R}^{3 \times 3}$ denotes the lattice matrix, every column vector of which is the periodic basic vector of the crystal. We can get the Cartesian coordinates representation of unit cell's atom coordinates $\boldsymbol{X}$ by $\boldsymbol{X} = \boldsymbol{L}\boldsymbol{F} \in \mathbb{R}^{3 \times N}$. The ideal infinite periodic crystal structure of $\mathcal{M}$ can be represented by $\{(a_i', \boldsymbol{x}_i') | a_i' = a_i, \boldsymbol{x}_i' = \boldsymbol{x}_i + \boldsymbol{L}\boldsymbol{k}, \forall \boldsymbol{k} \in \mathbb{Z}^{3 \times 1}\}$. Based on the above notations, the symmetry of crystal geometry is defined as *periodic E(3) invariance*[2] (Jiao et al., 2023), including periodic translational invariance of $\boldsymbol{F}$ and rotational invariance of $\boldsymbol{L}$ (details in Appendix B) .

**Bayesian Flow Networks** Different from the well-established SDE-based approaches, *e.g.* Diffusion Models (Ho et al., 2020a; Song et al., 2020b;a) and ODE-based approaches *e.g.* Flow Matching (Lipman et al., 2022), Bayesian Flow Networks define a generative process driven by consecutive Bayesian updates on noised samples from the uninformative prior distribution $\boldsymbol{\theta}_0$ to posteriors $\boldsymbol{\theta}_i$ with higher confidence and more information.

To define the consecutive Bayesian update process, Bayesian Flow Networks contain a process to add noise to the clean data samples which is an analogy to the forward process in Diffusion

---

[1]For a more detailed discussion of related work, please refer to Appendix F.

[2]Actually some crystal structures are chiral (Flack, 2003) and their structures are SE(3) invariant. In this paper, we follow Jiao et al. (2023) to build an E(3) invariant framework for fair comparison, while the SE(3) equivariance can be simply achieved by transforming an E(3) invariant network into an SE(3) equivariant network by excluding invariance to reflections in the Markov chain.

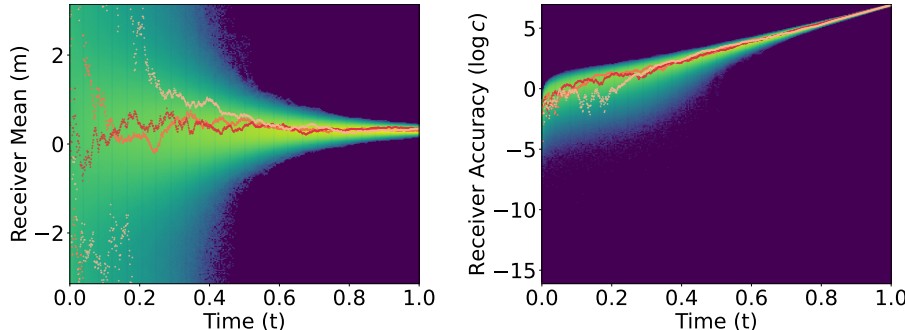

Figure 2: **Visualization of the proposed periodic Bayesian flow with mean parameter $\mu$ and accumulated accuracy parameter $c$ which corresponds to the entropy/uncertainty**. For $x = 0.3, \beta(1) = 1000$ and $\alpha_i$ defined in Appendix A, this figure plots three colored stochastic parameter trajectories for receiver mean parameter $m$ and accumulated accuracy parameter $c$, superimposed on a log-scale heatmap of the Bayesian flow distribution $p_F(m|x, \alpha_1, \alpha_2, \ldots, \alpha_i)$ and $p_F(c|x, \alpha_1, \alpha_2, \ldots, \alpha_i)$. Note the *non-monotonicity* and *non-additive* property of $c$ which could inform the network the entropy of the mean parameter $m$ as a condition and the *periodicity* of $m$.

**Models.** In BFN, such a process is explicitly defined by the so-called *sender distribution* $p_S(\mathbf{y}|\mathbf{x}; \alpha)$ where $\alpha$ is the parameter of sender distribution which corresponds to noise level, *e.g.* variance for Gaussian-formed $p_S$.

BFN aims to create a procedure that gradually acquires information from the ground truth data $\mathbf{x}$ to provide a training signal. To this end, the framework will sample a series of noisy samples $\mathbf{y}_1, \mathbf{y}_2, ..., \mathbf{y}_n$ independently from the *sender distribution* $p_S$ with various accuracy levels $\alpha_1, \alpha_2, ..., \alpha_n$. Based on samples, the framework would simulate an auto-regressive update of $\boldsymbol{\theta}$ based on the *Bayesian update function* to reflect an information gathering process from prior to data as:

$$\boldsymbol{\theta}_i = h(\boldsymbol{\theta}_{i-1}, \mathbf{y}_i, \alpha_i) \tag{3}$$

The neural network $\Psi$ is trained in a teacher-forcing fashion: approximate the sender distribution $p_S(\cdot|\mathbf{x}; \alpha_i)$ generating $\mathbf{y}_i$ according to $\boldsymbol{\theta}_{i-1}$, which has integrated the information of $(\mathbf{y}_1, \cdots, \mathbf{y}_{i-1})$ through Eq. (3). The network $\Psi$ hence takes $\boldsymbol{\theta}$ obtained by Bayesian update as input, and the distribution implied by $p_I(\cdot|\boldsymbol{\theta})$ is termed *input distribution*. The network output $\Psi(\boldsymbol{\theta})$ is interpreted as the parameter of an updated distribution $p_O(\cdot|\Psi(\boldsymbol{\theta}))$ over sample space, referred to as *output distribution*. Combining the network output $\Psi(\boldsymbol{\theta})$ and the sender distribution, we obtain the approximation to $p_S(\mathbf{y}_i \mid \mathbf{x}; \alpha_i)$ as:

$$p_R(\mathbf{y}_i \mid \boldsymbol{\theta}_{i-1}, \Psi, \alpha_i) = \mathbb{E}_{p_O(\mathbf{x}'|\Psi(\boldsymbol{\theta}_{i-1}))} p_S(\mathbf{y}_i \mid \mathbf{x}'; \alpha_i) \tag{4}$$

Such distribution is named as *receiver distribution* $p_R$. Note that $\boldsymbol{\theta}_{i-1}$ can be seen as a deterministic function mapping, given $(\mathbf{y}_1, \cdots, \mathbf{y}_{i-1})$ as $\boldsymbol{\theta}_{i-1} = f(\mathbf{y}_1, \cdots, \mathbf{y}_{i-1})$ based on the deterministic update function in Eq. (3). By combining the objectives of different timesteps and taking expectation over trajectories, we obtain the training objective of BFN as:

$$\mathcal{L}(\Psi) = \mathbb{E}_{\mathbf{x} \sim p_{\text{data}}} \mathbb{E}_{\Pi_{i=1}^n p_S(\mathbf{y}_i|\mathbf{x}, \alpha_i)} D_{KL}(p_S(\mathbf{y}_i|\mathbf{x}, \alpha_i) || p_R(\mathbf{y}_i \mid \boldsymbol{\theta}_{i-1}, \Psi, \alpha_i)) \tag{5}$$

Here $p_{\text{data}}$ is the empirical distribution. Detailed derivation and discussion are provided in Appendix A. Additionally, we provide toy examples with minimal components illustrating how BFNs work in our code repository.

## 4 METHOD

In this section, we explain the detailed design of CrysBFN tackling theoretical and practical challenges. First, we describe how to derive our new formulation of Bayesian Flow Networks over hyper-torus $\mathbb{T}^D$ from scratch. Next, we illustrate the two key differences between CrysBFN and the original form of BFN: 1) a meticulously designed novel base distribution with different Bayesian update rules; and 2) different properties over the accuracy scheduling resulted from the periodicity and the new Bayesian update rules. Then, we present in detail the overall framework of CrysBFN over each manifold of the crystal space (*i.e.* fractional coordinates, lattice vectors, atom types) respecting *periodic E(3) invariance*.

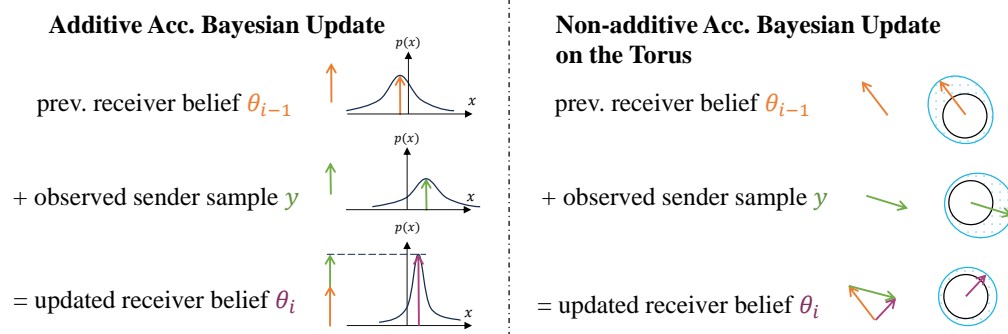

Figure 3: An intuitive illustration of non-additive accuracy Bayesian update on the torus. The lengths of arrows represent the uncertainty/entropy of the belief (*e.g.* $1/\sigma^2$ for Gaussian and $c$ for von Mises). The directions of the arrows represent the believed location (*e.g.* $\mu$ for Gaussian and $m$ for von Mises).

## 4.1 PERIODIC BAYESIAN FLOW ON HYPER-TORUS $\mathbb{T}^D$

For generative modeling of fractional coordinates in crystal, we first construct a periodic Bayesian flow on $\mathbb{T}^D$ by designing every component of the totally new Bayesian update process which we demonstrate to be distinct from the original Bayesian flow (please see Fig. 3).

The fractional atom coordinate system (Jiao et al., 2023) inherently distributes over a hyper-torus support $\mathbb{T}^{3 \times N}$. Hence, the normal distribution support on $\mathbb{R}$ used in the original (Graves et al., 2023) is not suitable for this scenario.

To tackle this problem, the circular distribution (Mardia & Jupp, 2009) over the finite interval $[-\pi, \pi)$ is a natural choice as the base distribution for deriving the BFN on $\mathbb{T}^D$. Specifically, circular distributions enjoy desirable periodic properties: 1) the integration over any interval length of $2\pi$ equals 1; 2) the probability distribution function is periodic with period $2\pi$. Sharing the same intrinsic with fractional coordinates, such periodic property of circular distribution makes it suitable for the instantiation of BFN's input distribution, in parameterizing the belief towards ground truth $\mathbf{x}$ on $\mathbb{T}^D$.

**von Mises Distribution and its Bayesian Update** We choose von Mises distribution (Mardia & Jupp, 2009) from various circular distributions as the form of input distribution, based on the appealing conjugacy property required in the derivation of the BFN framework. That is, the posterior of a von Mises distribution parameterized likelihood is still in the family of von Mises distributions. The probability density function of von Mises distribution with mean direction parameter $m$ and concentration parameter $c$ (describing the entropy/uncertainty of $m$) is defined as:

$$f(x|m, c) = vM(x|m, c) = \frac{\exp(c \cos(x - m))}{2\pi I_0(c)} \tag{6}$$

where $I_0(c)$ is zeroth order modified Bessel function of the first kind as the normalizing constant. Given the last univariate belief parameterized by von Mises distribution with parameter $\theta_{i-1} = \{m_{i-1}, c_{i-1}\}$ and the sample $y$ from sender distribution with unknown data sample $x$ and known accuracy $\alpha$ describing the entropy/uncertainty of $y$, Bayesian update for the receiver is deducted as:

$$h(\{m_{i-1}, c_{i-1}\}, y, \alpha) = \{m_i, c_i\}, \text{where} \tag{7}$$

$$m_i = \text{atan2}(\alpha \sin y + c_{i-1} \sin m_{i-1}, \alpha \cos y + c_{i-1} \cos m_{i-1}) \tag{8}$$

$$c_i = \sqrt{\alpha^2 + c_{i-1}^2 + 2\alpha c_{i-1} \cos(y - m_{i-1})} \tag{9}$$

The proof of the above equations can be found in Appendix A.3. The atan2 function refers to 2-argument arctangent. Independently conducting Bayesian update for each dimension, we can obtain the Bayesian update distribution by marginalizing $\mathbf{y}$:

$$p_U(\boldsymbol{\theta}'|\boldsymbol{\theta}, \mathbf{x}; \alpha) = \mathbb{E}_{p_S(\mathbf{y}|\mathbf{x}; \alpha)} \delta(\boldsymbol{\theta}' - h(\boldsymbol{\theta}, \mathbf{y}, \alpha)) = \mathbb{E}_{vM(\mathbf{y}|\mathbf{x}, \alpha)} \delta(\boldsymbol{\theta}' - h(\boldsymbol{\theta}, \mathbf{y}, \alpha)) \tag{10}$$

**Non-additive Accuracy** The additive accuracy is a nice property held with the Gaussian-formed sender distribution of the original BFN expressed as:

$$p_U(\boldsymbol{\theta}'' \mid \boldsymbol{\theta}, \mathbf{x}; \alpha_a + \alpha_b) = \mathbb{E}_{p_U(\boldsymbol{\theta}'|\boldsymbol{\theta}, \mathbf{x}; \alpha_a)} p_U(\boldsymbol{\theta}'' \mid \boldsymbol{\theta}', \mathbf{x}; \alpha_b) \tag{11}$$

Such property is mainly derived based on the standard identity of Gaussian variable:

$$X \sim \mathcal{N}\left(\mu_X, \sigma_X^2\right), Y \sim \mathcal{N}\left(\mu_Y, \sigma_Y^2\right) \implies X + Y \sim \mathcal{N}\left(\mu_X + \mu_Y, \sigma_X^2 + \sigma_Y^2\right) \quad (12)$$

The additive accuracy property makes it feasible to derive the Bayesian flow distribution $p_F(\boldsymbol{\theta} \mid \mathbf{x}; i) = p_U\left(\boldsymbol{\theta} \mid \boldsymbol{\theta}_0, \mathbf{x}, \sum_{k=1}^{i} \alpha_i\right)$ for the simulation-free training of Eq. (5). It should be noted that the standard identity in Eq. (11) does not hold in the von Mises distribution. Hence there exists an important difference between the original Bayesian flow defined on Euclidean space and the Bayesian flow of circular data on $\mathbb{T}^D$ based on von Mises distribution. With prior $\boldsymbol{\theta} = \{\mathbf{0}, \mathbf{0}\}$, we could formally represent the non-additive accuracy issue as:

$$p_U\left(c'' \mid \boldsymbol{\theta}, \mathbf{x}; \alpha_a + \alpha_b\right) = \delta(c - \alpha_a - \alpha_b) \neq \mathbb{E}_{p_U(\boldsymbol{\theta}'|\boldsymbol{\theta},\mathbf{x};\alpha_a)} p_U\left(c'' \mid \boldsymbol{\theta}', \mathbf{x}; \alpha_b\right)$$

$$= \mathbb{E}_{vM(\mathbf{y}_b|\mathbf{x},\alpha_a)} \mathbb{E}_{vM(\mathbf{y}_a|\mathbf{x},\alpha_b)} \delta(c - ||[\alpha_a \cos \mathbf{y}_a + \alpha_b \cos \mathbf{y}_b, \alpha_a \sin \mathbf{y}_a + \alpha_b \sin \mathbf{y}_b]^T||_2) \quad (13)$$

A more intuitive visualization could be found in Fig. 3. This fundamental difference between periodic Bayesian flow and that of Graves et al. (2023) presents both theoretical and practical challenges, which we will explain and address in the following contents.

**Entropy Conditioning** As a common practice in generative models (Ho et al., 2020a; Lipman et al., 2022; Graves et al., 2023), timestep $t$ is widely used to distinguish among generation states by feeding the timestep information into the networks. However, this paper shows that for periodic Bayesian flow, the accumulated accuracy $\boldsymbol{c}_i$ is more effective than time-based conditioning by informing the network about the entropy and certainty of the states $\boldsymbol{\theta}_i$. This stems from the intrinsic non-additive accuracy which makes the receiver's accumulated accuracy $c$ not bijective function of $t$, but a distribution conditioned on accumulated accuracies $\boldsymbol{c}_i$ instead. Therefore, the entropy parameter $\boldsymbol{c}$ is taken logarithm and fed into the network to describe the entropy of the input corrupted structure. We verify this consideration in Sec. 5.3.

**Reformulations of BFN**. Recall the original update function with Gaussian sender distribution, after receiving noisy samples $\mathbf{y}_1, \mathbf{y}_2, \ldots, \mathbf{y}_i$ with accuracies $\alpha_1, \alpha_2, \ldots, \alpha_i$, the accumulated accuracies of the receiver side could be analytically obtained by the additive property and it is consistent with the sender side. However, as previously mentioned, this does not apply to periodic Bayesian flow, and some of the notations in original BFN (Graves et al., 2023) need to be adjusted accordingly. We maintain the notations of sender side's one-step accuracy $\alpha$ and added accuracy $\beta$, and alter the notation of receiver's accuracy parameter as $c$, which is needed to be simulated by cascade of Bayesian updates. We emphasize that the receiver's accumulated accuracy $c$ is no longer a function of $t$ (differently from the Gaussian case), and it becomes a distribution conditioned on received accuracies $\alpha_1, \alpha_2, \ldots, \alpha_i$ from the sender. Therefore, we represent the Bayesian flow distribution of von Mises distribution as $p_F(\boldsymbol{\theta}|\mathbf{x}; \alpha_1, \alpha_2, \ldots, \alpha_i)$. And the original simulation-free training with Bayesian flow distribution is no longer applicable in this scenario.

**Fast Sampling from Equivalent Bayesian Flow Distribution** Based on the above reformulations, the Bayesian flow distribution of von Mises distribution is reframed as:

$$p_F(\boldsymbol{\theta}_i|\mathbf{x}; \alpha_1, \alpha_2, \ldots, \alpha_i) = \mathbb{E}_{p_U(\boldsymbol{\theta}_1|\boldsymbol{\theta}_0,\mathbf{x};\alpha_1)} \cdots \mathbb{E}_{p_U(\boldsymbol{\theta}_{i-1}|\boldsymbol{\theta}_{i-2},\mathbf{x};\alpha_{i-1})} p_U(\boldsymbol{\theta}_i|\boldsymbol{\theta}_{i-1},\mathbf{x}; \alpha_i) \quad (14)$$

Naively sampling from Eq. (14) requires slow auto-regressive iterated simulation, making training unaffordable. Noticing the mathematical properties of Eqs. (8) and (9), we transform Eq. (14) to the equivalent form:

$$p_F(\boldsymbol{m}_i|\mathbf{x}; \alpha_1, \alpha_2, \ldots, \alpha_i) = \mathbb{E}_{vM(\mathbf{y}_1|\mathbf{x},\alpha_1)\ldots vM(\mathbf{y}_i|\mathbf{x},\alpha_i)} \delta(\boldsymbol{m}_i - \mathrm{atan2}(\sum_{j=1}^{i} \alpha_j \cos \mathbf{y}_j, \sum_{j=1}^{i} \alpha_j \sin \mathbf{y}_j))$$

$$(15)$$

$$p_F(\boldsymbol{c}_i|\mathbf{x}; \alpha_1, \alpha_2, \ldots, \alpha_i) = \mathbb{E}_{vM(\mathbf{y}_1|\mathbf{x},\alpha_1)\ldots vM(\mathbf{y}_i|\mathbf{x},\alpha_i)} \delta(\boldsymbol{c}_i - ||[\sum_{j=1}^{i} \alpha_j \cos \mathbf{y}_j, \sum_{j=1}^{i} \alpha_j \sin \mathbf{y}_j]^T||_2)$$

$$(16)$$

which bypasses the computation of intermediate variables and allows pure tensor operations, with negligible computational overhead.

**Proposition 4.1.** *The probability density function of Bayesian flow distribution defined by Eqs. (15) and (16) is equivalent to the original definition in Eq. (14).*

**Numerical Determination of Linear Entropy Sender Accuracy Schedule** Original BFN designs the accuracy schedule $\beta(t)$ to make the entropy of input distribution linearly decrease. As for crystal generation task, to ensure information coherence between modalities, we choose a sender accuracy schedule $\alpha_1, \alpha_2, \ldots, \alpha_i$ that makes the receiver's belief entropy $H(t_i) = H(p_I(\cdot|\boldsymbol{\theta}_i)) = H(p_I(\cdot|\boldsymbol{c}_i))$ linearly decrease *w.r.t.* time $t_i$, given the initial and final accuracy parameter $c(0)$ and $c(1)$. Due to the intractability of Eq. (30), we first use numerical binary search in $[0, c(1)]$ to determine the receiver's $c(t_i)$ for $i = 1, \ldots, n$ by solving the equation $H(c(t_i)) = (1 - t_i)H(c(0)) + tH(c(1))$. Next, with $c(t_i)$, we conduct numerical binary search for each $\alpha_i$ in $[0, c(1)]$ by solving the equations $\mathbb{E}_{y \sim vM(x, \alpha_i)}[\sqrt{\alpha_i^2 + c_{i-1}^2 + 2\alpha_i c_{i-1} \cos(y - m_{i-1})}] = c(t_i)$ (by Eq. (45)) from $i = 1$ to $i = n$ for arbitrarily selected $x \in [-\pi, \pi)$.

After tackling all those issues, we have now arrived at a new BFN architecture for effectively modeling crystals. Such BFN can also be adapted to other type of data located in hyper-torus $\mathbb{T}^D$.

## 4.2 EQUIVARIANT BAYESIAN FLOW FOR CRYSTAL

With the above Bayesian flow designed for generative modeling of fractional coordinate $\boldsymbol{F}$, we are able to build equivariant Bayesian flow for each modality of crystal. In this section, we first give an overview of the general training and sampling algorithm of CrysBFN (visualized in Fig. 1). Then, we describe the details of the Bayesian flow of every modality. The training and sampling algorithm can be found in Algorithm 1 and Algorithm 2.

**Overview** Operating in the parameter space $\boldsymbol{\theta}^{\mathcal{M}} = \{\boldsymbol{\theta}^A, \boldsymbol{\theta}^L, \boldsymbol{\theta}^F\}$, CrysBFN generates high-fidelity crystals through a joint BFN sampling process on the parameter of atom type $\boldsymbol{\theta}^A$, lattice parameter $\boldsymbol{\theta}^L = \{\boldsymbol{\mu}^L, \boldsymbol{\rho}^L\}$, and the parameter of fractional coordinate matrix $\boldsymbol{\theta}^F = \{\boldsymbol{m}^F, \boldsymbol{c}^F\}$. We index the $n$-steps of the generation process in a discrete manner $i$, and denote the corresponding continuous notation $t_i = i/n$ from prior parameter $\boldsymbol{\theta}_0^{\mathcal{M}}$ to a considerably low variance parameter $\boldsymbol{\theta}_n^{\mathcal{M}}$ (*i.e.* large $\boldsymbol{\rho}^L, \boldsymbol{m}^F$, and centered $\boldsymbol{\theta}^A$).

At training time, CrysBFN samples time $i \sim U\{1, n\}$ and $\boldsymbol{\theta}_{i-1}^{\mathcal{M}}$ from the Bayesian flow distribution of each modality, serving as the input to the network. The network $\Psi$ outputs $\Psi(\boldsymbol{\theta}_{i-1}^{\mathcal{M}}, t_{i-1}) = \Psi(\boldsymbol{\theta}_{i-1}^A, \boldsymbol{\theta}_{i-1}^F, \boldsymbol{\theta}_{i-1}^L, t_{i-1})$ and conducts gradient descents on loss function Eq. (5) for each modality. After proper training, the sender distribution $p_S$ can be approximated by the receiver distribution $p_R$.

At inference time, from predefined $\boldsymbol{\theta}_0^{\mathcal{M}}$, we conduct transitions from $\boldsymbol{\theta}_{i-1}^{\mathcal{M}}$ to $\boldsymbol{\theta}_i^{\mathcal{M}}$ by: (1) sampling $\mathbf{y}_i \sim p_R(\mathbf{y}|\boldsymbol{\theta}_{i-1}^{\mathcal{M}}; t_i, \alpha_i)$ according to network prediction $\hat{\Psi}_M(\boldsymbol{\theta}_{i-1}^{\mathcal{M}}, t_{i-1})$; and (2) performing Bayesian update $h(\boldsymbol{\theta}_{i-1}^{\mathcal{M}}, \mathbf{y}_{i-1}^{\mathcal{M}}, \alpha_i)$ for each dimension.

**Bayesian Flow of Fractional Coordinate $\boldsymbol{F}$** The distribution of the prior parameter $\boldsymbol{\theta}_0^F$ is defined as:

$$p(\boldsymbol{\theta}_0^F) \stackrel{\text{def}}{=} \{vM(\boldsymbol{m}_0^F|\mathbf{0}_{3 \times N}, \mathbf{0}_{3 \times N}), \delta(\boldsymbol{c}_0^F - \mathbf{0}_{3 \times N})\} = \{U(\mathbf{0}, \mathbf{1}), \delta(\boldsymbol{c}_0^F - \mathbf{0}_{3 \times N})\} \quad (17)$$

Note that this prior distribution of $\boldsymbol{m}_0^F$ is uniform over $[\mathbf{0}, \mathbf{1}]$, ensuring the periodic translation invariance property in Definition 1. The training objective is minimizing the KL divergence between sender and receiver distribution (deduction can be found in Appendix A.7):

$$\mathcal{L}_F = n\mathbb{E}_{i \sim U\{1,n\}, p_F(\boldsymbol{\theta}^F|\boldsymbol{F}; \alpha_1, \alpha_2, \ldots, \alpha_i)} \alpha_i \frac{I_1(\alpha_i)}{I_0(\alpha_i)}(1 - \cos(\boldsymbol{F} - \hat{\Psi}_F(\boldsymbol{\theta}_{i-1}^{\mathcal{M}}, t_{i-1}))) \quad (18)$$

where $I_0(x)$ and $I_1(x)$ are the zeroth and the first order of modified Bessel functions. The transition from $\boldsymbol{\theta}_{i-1}^F$ to $\boldsymbol{\theta}_i^F$ is the Bayesian update distribution based on network prediction:

$$p(\boldsymbol{\theta}_i^F|\boldsymbol{\theta}_{i-1}^{\mathcal{M}}) = \mathbb{E}_{vM(\mathbf{y}|\hat{\Psi}_F(\boldsymbol{\theta}_{i-1}^{\mathcal{M}}, t_{i-1}), \alpha_i)} \delta(\boldsymbol{\theta}_i^F - h(\boldsymbol{\theta}_{i-1}^F, \mathbf{y}, \alpha_i)) \quad (19)$$

**Proposition 4.2.** *With $\Psi_F$ as a periodic translation equivariant function namely $\Psi_F(\boldsymbol{\theta}^A, w(\boldsymbol{\theta}^F + \boldsymbol{t}), \boldsymbol{\theta}^L, t) = w(\Psi_F(\boldsymbol{\theta}^A, \boldsymbol{\theta}^F, \boldsymbol{\theta}^L, t) + \boldsymbol{t}), \forall \boldsymbol{t} \in \mathbb{R}^3$, the marginal distribution of $p(\boldsymbol{F}_n)$ defined by Eqs.* (17) *and* (19) *is periodic translation invariant.*

**Bayesian Flow of Lattice Parameter $\boldsymbol{L}$** Noting the lattice parameter $\boldsymbol{L}$ located in Euclidean space, we set prior as the parameter of a isotropic multivariate normal distribution $\boldsymbol{\theta}_0^L \stackrel{\text{def}}{=} \{\boldsymbol{\mu}_0^L, \boldsymbol{\rho}_0^L\} =$

$\{\mathbf{0}_{3\times 3}, \mathbf{1}_{3\times 3}\}$ such that the prior distribution of the Markov process on $\boldsymbol{\mu}^L$ is the Dirac distribution $\delta(\boldsymbol{\mu_0} - \mathbf{0})$ and $\delta(\boldsymbol{\rho_0} - \mathbf{1})$, which ensures O(3)-invariance of prior distribution of $\boldsymbol{L}$. By Eq. 77 from Graves et al. (2023), the Bayesian flow distribution of the lattice parameter $\boldsymbol{L}$ is:

$$p_F^L(\boldsymbol{\mu}^L|\boldsymbol{L};t) = \mathcal{N}(\boldsymbol{\mu}^L|\gamma(t)\boldsymbol{L}, \gamma(t)(1-\gamma(t))\boldsymbol{I}) \tag{20}$$

where $\gamma(t) = 1 - \sigma_1^{2t}$ and $\sigma_1$ is the predefined hyper-parameter controlling the variance of input distribution at $t = 1$ under linear entropy accuracy schedule. The variance parameter $\boldsymbol{\rho}$ does not need to be modeled and fed to the network, since it is deterministic given the accuracy schedule. After sampling $\boldsymbol{\mu}_i^L$ from $p_F^L$, the training objective is defined as minimizing KL divergence between sender and receiver distribution (based on Eq. 96 in Graves et al. (2023)):

$$\mathcal{L}_L = \frac{n}{2}\left(1 - \sigma_1^{2/n}\right) \mathbb{E}_{i\sim U\{1,n\}} \mathbb{E}_{p_F(\boldsymbol{\mu}_{i-1}^L|\boldsymbol{L};t_{i-1})} \frac{\left\|\boldsymbol{L} - \hat{\Psi}_L(\boldsymbol{\theta}_{i-1}^{\mathcal{M}}, t_{i-1})\right\|^2}{\sigma_1^{2i/n}}, \tag{21}$$

where the prediction term $\hat{\Psi}_L(\boldsymbol{\theta}_{i-1}^{\mathcal{M}}, t_{i-1})$ is the lattice parameter part of network output. After training, the generation process is defined as the Bayesian update distribution given network prediction:

$$p(\boldsymbol{\mu}_i^L|\boldsymbol{\theta}_{i-1}^{\mathcal{M}}) = p_U^L(\boldsymbol{\mu}_i^L|\hat{\Psi}_L(\boldsymbol{\theta}_{i-1}^{\mathcal{M}}, t_{i-1}), \boldsymbol{\mu}_{i-1}^L; t_{i-1}) \tag{22}$$

**Proposition 4.3.** *With $\Psi_L$ as O(3)-equivariant function namely $\Psi_L(\boldsymbol{\theta}^A, \boldsymbol{\theta}^F, \boldsymbol{Q}\boldsymbol{\theta}^L, t) = \boldsymbol{Q}\Psi_L(\boldsymbol{\theta}^A, \boldsymbol{\theta}^F, \boldsymbol{\theta}^L, t), \forall \boldsymbol{Q}^T\boldsymbol{Q} = \boldsymbol{I}$, the marginal distribution of $p(\boldsymbol{\mu}_n^L)$ defined by Eq. (22) is O(3)-invariant.*

**Bayesian Flow of Atom Types $A$** Given that atom types are discrete random variables located in a simplex $\mathcal{S}^K$, the prior parameter of $\boldsymbol{A}$ is the discrete uniform distribution over the vocabulary $\boldsymbol{\theta}_0^A \overset{\text{def}}{=} \frac{1}{K}\mathbf{1}_{1\times N}$. With the notation of the projection from the class index $j$ to the length $K$ one-hot vector $(\mathbf{e}_j)_k \overset{\text{def}}{=} \delta_{jk}$, where $\mathbf{e}_j \in \mathbb{R}^K$, $\mathbf{e}_{\boldsymbol{A}} \overset{\text{def}}{=} (\mathbf{e}_{a_1}, \ldots, \mathbf{e}_{a_N}) \in \mathbb{R}^{K\times N}$, the Bayesian flow distribution of atom types $\boldsymbol{A}$ is derived in Graves et al. (2023):

$$p_F^A(\boldsymbol{\theta}^A \mid \boldsymbol{A}; t) = \mathbb{E}_{\mathcal{N}(\mathbf{y}|\beta^A(t)(K\mathbf{e}_{\boldsymbol{A}}-\mathbf{1}_{K\times N}), \beta^A(t)K\boldsymbol{I}_{K\times N\times N})} \delta\left(\boldsymbol{\theta}^A - \frac{e^{\mathbf{y}}\boldsymbol{\theta}_0^A}{\sum_{k=1}^K e^{\mathbf{y}_k}(\boldsymbol{\theta}_0)_k^A}\right). \tag{23}$$

where $\beta^A(t)$ is the predefined accuracy schedule for atom types. Sampling $\boldsymbol{\theta}_i^A$ from $p_F^A$ as the training signal, the training objective is the $n$-step discrete-time loss for discrete variable (Graves et al., 2023):

$$\mathcal{L}_A = n\mathbb{E}_{i\sim U\{1,n\}, p_F^A(\boldsymbol{\theta}^A|\boldsymbol{A};t_{i-1}), \mathcal{N}(\mathbf{y}|\alpha_i(K\mathbf{e}_{\boldsymbol{A}}-\mathbf{1}), \alpha_i K\boldsymbol{I})} \ln \mathcal{N}\left(\mathbf{y} \mid \alpha_i(K\mathbf{e}_{\boldsymbol{A}} - \mathbf{1}), \alpha_i K\boldsymbol{I}\right)$$

$$- \sum_{d=1}^N \ln\left(\sum_{k=1}^K p_O^{(d)}(k \mid \boldsymbol{\theta}^A; t_{i-1})\mathcal{N}\left(y^{(d)} \mid \alpha_i(K\mathbf{e}_k - \mathbf{1}), \alpha_i K\boldsymbol{I}\right)\right) \tag{24}$$

where $\boldsymbol{I} \in \mathbb{R}^{K\times N\times N}$ and $\mathbf{1} \in \mathbb{R}^{K\times D}$. When sampling, the transition from $\boldsymbol{\theta}_{i-1}^A$ to $\boldsymbol{\theta}_i^A$ is derived as:

$$p(\boldsymbol{\theta}_i^A|\boldsymbol{\theta}_{i-1}^{\mathcal{M}}) = p_U^A(\boldsymbol{\theta}_i^A|\boldsymbol{\theta}_{i-1}^A, \hat{\Psi}_A(\boldsymbol{\theta}_{i-1}^{\mathcal{M}}, t_{i-1}); t_{i-1}) \tag{25}$$

The detailed training and sampling algorithm could be found in Algorithm 1 and Algorithm 2.

## 5 EXPERIMENTS

We evaluate on two crystal generation tasks: ab initio generation in Sec. 5.1 and stable structure prediction task in Sec. 5.2. Ablation studies are detailed in Sec. 5.3 to validate design choices. We provide the implementation details in Appendix C.

Following Xie et al. (2021); Jiao et al. (2023), we choose the following datasets for evaluation: 1) **Perov-5** (Castelli et al., 2012a;b) is composed of 18,928 perovskite crystals of similar structures, with 5 atoms in a unit cell sharing the chemical formula $ABX_3$. 2) **Carbon-24** (Pickard, 2020) contains 10,153 crystals with 6~24 atoms in a cell, and all atom types are carbon. 3) **MP-20** (Jain et al., 2013) selects 45,231 stable inorganic materials from Material Projects (Jain et al., 2013), including the majority of experimentally-verified materials with at most 20 atoms in a unit cell. 4) **MPTS-52** (Jiao et al., 2023) consists of 40,476 crystals up to 52 atoms per cell, which is a more challenging extension of MP-20. All crystals are reduced as Niggli cells (Niggli, 1928). The procedure to split the datasets into training, validation, and testing subsets adheres to prior practices (Xie et al., 2021; Jiao et al., 2023).

Table 1: Results on ab initio generation task. Baseline results are from Xie et al. (2021); Jiao et al. (2023); Miller et al. (2024).

| Data | Method | Validity (%) ↑ | | Coverage (%) ↑ | | Property ↓ | | |
|------|--------|------|------|------|------|------|------|------|
| | | Struc. | Comp. | COV-R | COV-P | $d_\rho$ | $d_E$ | $d_{elem}$ |
| Perov-5 | Cond-DFC-VAE (Court et al., 2020) | 73.60 | 82.95 | 73.92 | 10.13 | 2.268 | 4.111 | 0.8373 |
| | G-SchNet (Gebauer et al., 2019) | 99.92 | 98.79 | 0.18 | 0.23 | 1.625 | 4.746 | 0.0368 |
| | P-G-SchNet (Gebauer et al., 2019) | 79.63 | **99.13** | 0.37 | 0.25 | 0.2755 | 1.388 | 0.4552 |
| | CDVAE (Xie et al., 2021) | **100.0** | 98.59 | 99.45 | 98.46 | 0.1258 | 0.0264 | 0.0628 |
| | DiffCSP(Jiao et al., 2023) | **100.0** | 98.85 | **99.74** | 98.27 | 0.1110 | 0.0263 | 0.0128 |
| | CrysBFN | **100.0** | 98.86 | 99.52 | **98.63** | **0.0728** | **0.0198** | **0.0098** |
| Carbon-24 | G-SchNet (Gebauer et al., 2019) | 99.94 | – | 0.00 | 0.00 | 0.9427 | 1.320 | – |
| | P-G-SchNet (Gebauer et al., 2019) | 48.39 | – | 0.00 | 0.00 | 1.533 | 134.7 | – |
| | CDVAE (Xie et al., 2021) | **100.0** | – | 99.80 | 83.08 | 0.1407 | 0.2850 | – |
| | DiffCSP (Jiao et al., 2023) | **100.0** | – | **99.90** | 97.27 | 0.0805 | 0.0820 | – |
| | CrysBFN | **100.0** | – | **99.90** | **99.12** | **0.0612** | **0.0503** | – |
| MP-20 | G-SchNet (Gebauer et al., 2019) | 99.65 | 75.96 | 38.33 | 99.57 | 3.034 | 42.09 | 0.6411 |
| | P-G-SchNet (Gebauer et al., 2019) | 77.51 | 76.40 | 41.93 | 99.74 | 4.04 | 2.448 | 0.6234 |
| | CDVAE (Xie et al., 2021) | **100.0** | 86.70 | 99.15 | 99.49 | 0.6875 | 0.2778 | 1.432 |
| | DiffCSP(Jiao et al., 2023) | **100.0** | 83.25 | **99.71** | 99.76 | 0.3502 | 0.1247 | 0.3398 |
| | FlowMM (Miller et al., 2024) | 96.85 | 83.19 | 99.49 | 99.58 | 0.239 | - | **0.083** |
| | CrysBFN | **100.0** | **87.51** | 99.09 | **99.79** | **0.2067** | **0.0632** | 0.1628 |

Table 2: Results on stable structure prediction task. Baseline results are from Jiao et al. (2023); Miller et al. (2024).

| | Perov-5 | | MP-20 | | MPTS-52 | |
|------|------|------|------|------|------|------|
| | Match rate↑ | RMSE↓ | Match rate↑ | RMSE↓ | Match rate↑ | RMSE↓ |
| CDVAE (Xie et al., 2021) | 45.31 | 0.1138 | 33.90 | 0.1045 | 5.34 | 0.2106 |
| DiffCSP (Jiao et al., 2023) | 52.02 | 0.0760 | 51.49 | 0.0631 | 12.19 | 0.1786 |
| FlowMM (Miller et al., 2024) | 53.15 | 0.0992 | 61.39 | 0.0566 | 17.54 | 0.1726 |
| CrysBFN | **54.69** | **0.0636** | **64.35** | **0.0433** | **20.52** | **0.1038** |

## 5.1 AB INITIO GENERATION

**Baselines** For this task, the compared baselines include: 2) two-stage VAE-based methods **Cond-DFC-VAE** (Court et al., 2020) and **CDVAE** (Xie et al., 2021); 2) auto-regressive method **G-SchNet** (Gebauer et al., 2019), and its periodic adaptation **P-G-SchNet** (Xie et al., 2021); 3) diffusion-based joint generation approach **DiffCSP** (Jiao et al., 2023). 4) flow-matching-based approach **FlowMM** (Note that FlowMM only reports results on MP-20 and excludes $d_E$). We follow (Hoogeboom et al., 2022; Jiao et al., 2023) to sample atom numbers from a distribution that is pre-computed based on atom numbers in the training dataset. **Performance Indicators** Following previous work (Xie et al., 2021), we evaluate the efficacy of our model from three aspects: 1) *Validity*: Structure and compositional validity of randomly generated 10000 materials. 2) *Coverage*: Coverage score between generated 10000 materials and test set, defined by average minimum structure distance and average minimum compositional distance. 3) *Property Statistics*: the earth mover's distance (EMD) between the property distribution of generated crystals and test dataset crystals. Monitored properties include density ($\rho$, unit g/cm$^3$), energy predicted by an independent GNN ($E$, unit eV/atom), and the number of unique elements (# elem.).

**Results** The evaluation metrics for ab initio generation tasks are listed in Tab. 1. Our method consistently achieves better or competitive property statistics and generation precision on three datasets compared to baseline generative models. For compositional metrics including $d_{elem}$ and compositional validity, our method demonstrates bigger performance improvement for the more challenging dataset MP-20 (+4.34% compared to DiffCSP with the same level of $d_{elem}$), underscoring the importance of modeling atom types in the simplex space.

## 5.2 STABLE STRUCTURE PREDICTION

In this section, we extend our method to stable structure prediction task, where the modeling target is $p(\boldsymbol{L}, \boldsymbol{F}|\boldsymbol{A})$. The condition of atom types $\boldsymbol{A}$ is incorporated into the network by concatenating node feature and atom type embedding, following Jiao et al. (2023).

**Baselines** Following the practices in Jiao et al. (2023), we select baselines generative approaches including Diffusion-based approaches **CDVAE** and **DiffCSP** and the recent flow-matching-based method, **FlowMM**, which only reports results on the MP-20 dataset for this task. **Performance Indicators** The measured performance indicators for this task are **Match Rate** and **RMSD** computed by `StructureMatcher` class with thresholds `stol=0.5`, `angle_tol=10`, `ltol=0.3` in pymatgen (Ong et al., 2013), between the predicted structure candidates and the groundtruth structure given the composition. **Results** As summarized in Tab. 2, CrysBFN achieves consistent performance improvement over baseline methods, especially for more challenging datasets ($\sim 13\%$ higher match rate than DiffCSP for MP-20 and $\sim 40\%$ lower RMSE compared to FlowMM for MPTS-52).

## 5.3 ABLATION STUDY

Using MP-20 dataset and stable structure prediction task, we validate the necessity of proposed components of CrysBFN with results summarized in Tab. 3:

1) By removing the entropy parameter condition $c^F$ and using time as condition, match rate drops to $52.16\%$, proving that, different from original BFN, the non-additive accuracy dynamics requires the network to model both mean parameter $m$ and entropy parameter condition $c$.

2) By altering the approximated linear-entropy sender accuracy schedule to the hand-designed roughly linear-entropy schedule $c(t) = tc^t(1)$, we validate the effect of exact searched linear entropy schedule.

Table 3: Ablation studies on MP-20.

|  | Match rate (%) ↑ | RMSE ↓ |
|---|---|---|
| CrysBFN | **64.35** | **0.0433** |
| $w/o$ entropy cond. | 52.16 | 0.0631 |
| $w/o$ approx. sch. | 49.76 | 0.0643 |
| $w/o$ torus BFN | 6.17 | 0.3822 |
| | 1k Batches Sim. Time (s) | |
| Iterated Sim. | 356.1 | |
| Fast Sim. | 92.6 | |

3) By replacing the proposed hyper-torus BFN with the original continuous BFN, we observe poor match rate at $6.17\%$, indicating the importance of redesigning BFN for crystal data. Calculating the computational time for simulating 1000 batches, we observe $\sim 4\times$ efficiency, verifying fast sampling rate considering the full training procedure of MP-20 requires $\sim 150k$ steps.

## 5.4 SAMPLING EFFICIENCY EXPERIMENT

We compare the sampling efficiency of Crys-BFN and DiffCSP over the CSP task on the MP-20 dataset, based on the Number of Function Evaluations (NFE), *i.e.*, the number of network forward passes. The experimental results is plotted in Fig. 4. Notably, CrysBFN achieves a remarkable match rate of $60.02\%$ with only 10 step network forwards, surpassing DiffCSP's performance of $51.49\%$ at 2000 step network forwards. This illustrates the exceptional sampling efficiency of CrysBFN.

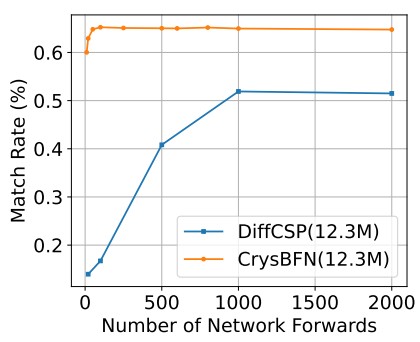

Figure 4: Experimental results on MP-20 with different Number of Function Evaluations (NFE) *i.e.* number of network forward passes.

## 6 CONCLUSION

In this paper, we present the first periodic Bayesian flow modeling on the hyper-torus, addressing an unprecedented theoretical issue related to non-additive accuracy. Specifically, we introduce a novel entropy conditioning mechanism, theoretical reformulations of BFN, a fast sampling algorithm, and a numerical method for determining the accuracy schedule. Leveraging the proposed periodic Bayesian flow, we implement the first periodic E(3) equivariant Bayesian flow networks for crystal generation. Our approach achieves state-of-the-art performance in crystal generation, with efficiency improved by two orders of magnitude. Additionally, our methodology can be adapted to a wide range of data types and tasks involving hyper-torus data.

ETHICS STATEMENT

We confirm that our work complies with the ICLR Code of Ethics, and we have carefully considered potential ethical concerns related to the development and use of our proposed method, CrysBFN, for crystal generation. Our method is designed for general crystal generative modeling tasks and does not involve sensitive data or tasks. We strongly encourage users to ensure compliance with relevant privacy regulations and critically assess the model's outputs. We are confirmed that there is no conflict of interest, financial sources or other factors, that could influence the development or presentation of this work.

With these considerations, we do not anticipate any violations of the ICLR Code of Ethics in the development or use of this model. And we stress once again that CrysBFN should not be used for malicious purposes, such as creating harmful structures.

REPRODUCIBILITY STATEMENT

To ensure reproducibility, we give a detailed derivation of the periodic Bayesian flow in Appendix A and the proof of the propositions in Appendix B. All datasets and performance evaluation method used in our experiments are publicly available and clearly specified or cited in Sec. 5. We provide our implementation details including training and sampling procedure, hyper-parameters, used computational resources and anonymous code repository link in Appendix C.

ACKNOWLEDGMENTS

This work is supported by the National Science and Technology Major Project (2022ZD0117502), the Natural Science Foundation of China (Grant No. 62376133, 62406170) and sponsored by Beijing Nova Program (20240484682) and the Wuxi Research Institute of Applied Technologies, Tsinghua University under Grant 20242001120..

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

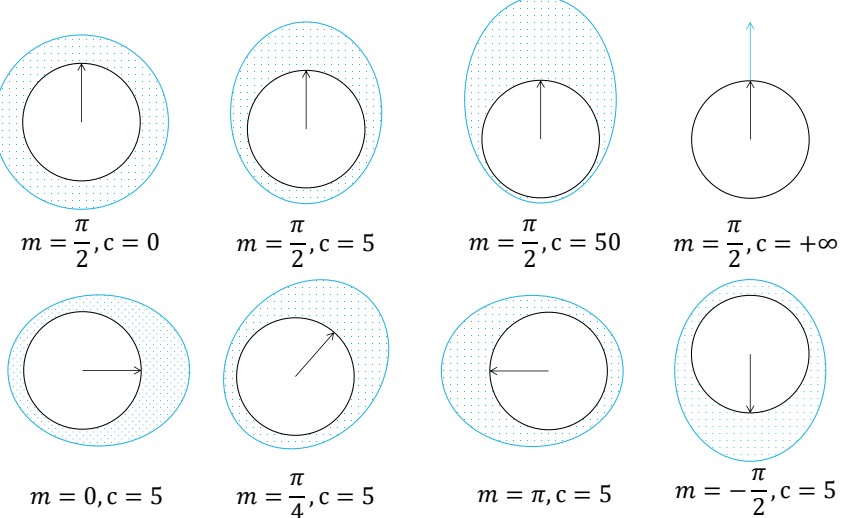

Figure 5: Depiction of von Mises distributions with different directions parameters $m$ and concentration parameters $c$. The parameter $m$ denotes the central location about which the distribution is centered, while $c$ functions as a measure of the distribution's concentration. When $c = 0$, the distribution is uniform on the circle. As $c$ increases, the distribution becomes more concentrated around the value $m$, with $c$ serving as a measure of this concentration. In the limit as $c \to +\infty$, the distribution converges to $\delta(m)$, a Dirac delta distribution centered at $m$.

## A BAYESIAN FLOW NETWORKS FOR CIRCULAR DATA

In this section, we provide a detailed derivation of Bayesian flow networks considering periodicity.

### A.1 CIRCULAR DATA AND VON MISES DISTRIBUTION

One-dimensional circular data $x$ refers to observations of random variables supporting on the circumference of the unit circle defined in directional statistics (Mardia & Jupp, 2009; Ley & Verdebout, 2017). This space can be represented by the torus:

$$\mathbb{T}^1 \overset{\text{def}}{=} \{\boldsymbol{z} \in \mathbb{R}^2 : ||\boldsymbol{z}|| = 1\} \tag{26}$$

For $n$-dimensional data $\boldsymbol{x} \in \mathbb{R}^n$, the set of $\boldsymbol{x}$ with every dimension located in $\mathbb{T}^1$ form a compact Riemannian manifold named hyper-torus $\mathbb{T}^n$ formally.

Wrapped normal distribution used in Jiao et al. (2023) and von Mises distribution used in this paper are both circular distributions defined in this space, the probability density function of von Mises distribution with mean direction parameter $m$ and concentration parameter $c$ is

$$f(x|m,c) = vM(x|m,c) = \frac{\exp(c\cos(x-m))}{2\pi I_0(c)} \tag{27}$$

where $I_0(c)$ is the modified Bessel function of the first kind of order 0 as the normalizing constant. The parameters $m$ and $1/c$ are analogous to mean $\mu$ and variance $\sigma^2$ in the normal distribution: 1) $m$ represents the central location around which the distribution is clustered, while $c$ serves as a measure of concentration. 2) We give a depiction of von Mises distributions with different directions parameters in Fig. 5. When $c$ equals zero, the distribution is uniform. As $c$ becomes large, the distribution becomes tightly concentrated around the value $m$, with $c$ quantifying this concentration. In the limit as $c \to +\infty$, the distribution becomes a Dirac delta distribution centered at $m$. Its support can be chosen as any interval of length $2\pi$ and we in this paper choose $[-\pi, \pi)$. Note that the fractional coordinate can be transformed to this interval easily by a linear transformation $g(x) = 2\pi x - \pi$. For this modeled interval, the map function from $\mathbb{R}$ to $[-\pi, \pi)$ is

$$w_{[-\pi,\pi)}(x) = (x - \pi)\%2\pi - \pi = x + 2\pi k, \exists k \in \mathbb{Z} \tag{28}$$

Such map function is equivalent to the map function $w(x) = x - \lfloor x \rfloor = x + k, \exists k \in \mathbb{Z}$ used in Jiao et al. (2023) if we choose the modelled interval as $[0, 1)$. Then, we can prove that the probability density distribution of von Mises distribution is equivariant to the periodic translation transformation:

$$
\begin{aligned}
\forall t \in \mathbb{R}, \quad f(w_{[-\pi,\pi)}(x+t)|w_{[-\pi,\pi)}(m+t), c) &= f(x + 2\pi k'|m + 2\pi k + t, c) \\
&= \frac{\exp(c\cos(x + 2\pi k' - (m + 2\pi k + t)))}{2\pi I_0(c)} \\
&= \frac{\exp(c\cos((x - m))}{2\pi I_0(c)} \\
&= f(x|m, c)
\end{aligned}
\tag{29}
$$

The differential entropy of von Mises distribution with mean direction parameter $m$ and concentration parameter $c$ is

$$
H(vM(x|m, c)) = -c\frac{I_1(c)}{I_0(c)} + \ln[2\pi I_0(c)] \quad \text{(Mardia \& Jupp, 2009)} \tag{30}
$$

We opt for von Mises distribution rather than the wrapped normal distribution used in Jiao et al. (2023); Jing et al. (2022) mainly because of the Bayesian conjugacy of von Mises distribution the posterior of which is conjugate to the prior distribution if the likelihood is parameterized as von Mises distribution which is the fundamental basis of constructing a Bayesian flow. Interestingly, there is an intriguing connection between von Mises distribution and crystal force field that the von Mises distribution is the stationary distribution of a drift and diffusion process on the circle in a harmonic potential corresponding to the harmonic force field of crystals (Risken & Risken, 1996).

## A.2 INPUT DISTRIBUTION $p_I(\,\cdot\,|\boldsymbol{\theta})$ AND SENDER DISTRIBUTION $p_S(\boldsymbol{y}|\boldsymbol{x}; \alpha)$

For circular data $\mathbf{x}$ which locates in a quotient space $\mathbb{T}^n = \mathbb{R}^{3\times N}/[-\pi, \pi)^{3\times N}$, we define the input distribution of the Bayesian Flow Networks as independently factorized von Mises distribution over the interval $[-\pi, \pi)$.

$$
\boldsymbol{\theta} \overset{\text{def}}{=} \{\mathbf{m}, \mathbf{c}\} \tag{31}
$$

$$
p_I(\mathbf{x}|\theta) \overset{\text{def}}{=} \Pi_{d=1}^D vM(x^{(d)}|m^{(d)}, c^{(d)}) \tag{32}
$$

where $m^{(d)} \in [-\pi, \pi)$ and $c^{(d)} \in [0, \infty)$.

In this paper, to ensuring the periodic translational invariance, the prior parameter of CrysBFN's Bayesian flow is chosed as

$$
p(\boldsymbol{\theta}_0^F) \overset{\text{def}}{=} \{vM(\boldsymbol{m}_0|\mathbf{0}_{3\times N}, \mathbf{0}_{3\times N}), \delta(\boldsymbol{c}_0 - \mathbf{0}_{3\times N})\} = \{U(\mathbf{0}, \mathbf{1}), \delta(\boldsymbol{c}_0 - \mathbf{0}_{3\times N})\} \tag{33}
$$

where $\mathbf{0}$ is the length of $D$ vector whose entries are all $0$. Note that this input prior $\boldsymbol{\theta}_0$ defines a multivariate uniform distribution

$$
p_I(\mathbf{x}|\boldsymbol{\theta}_0) = \Pi_{d=1}^D vM(x^{(d)}|0, 0) = \Pi_{d=1}^D U(-\pi, \pi) \tag{34}
$$

which ensures *periodic E(3) invariance* of the prior distribution.

The sender space $\mathcal{Y}$ is identical to the data space $\mathcal{X}$ for circular data. And the sender distribution is von Mises distribution centered on $\boldsymbol{x}$ with concentration parameter $\alpha$ represented by

$$
p_S(\mathbf{y}|\mathbf{x}; \alpha) = \Pi_{d=1}^D vM(y^{(d)}|x^{(d)}, \alpha) := vM(\mathbf{y}|\mathbf{x}, \alpha) \tag{35}
$$

## A.3 BAYESIAN UPDATE FUNCTION $h(\boldsymbol{\theta}_{i-1}, \mathbf{y}, \alpha)$ AND BAYESIAN UPDATE DISTRIBUTION $p_U(\cdot \mid \boldsymbol{\theta}, \mathbf{x}; \alpha)$

For the receiver, given his last univariate belief parameterized by von Mises distribution with parameter $\theta_{i-1} = \{m_{i-1}, c_{i-1}\}$, he now observes a sample $y$ from sender distribution with unknown

$x$ and known $\alpha$. Now, by Bayesian theorem,

$$p(x|y; \alpha, m_{i-1}, c_{i-1}) = \frac{p(y|x; \alpha)p(x; m_{i-1}, c_{i-1})}{p(y)} \tag{36}$$

$$\propto p(y|x; \alpha)p(x; m_{i-1}, c_{i-1}) \tag{37}$$

$$= vM(y|x, \alpha)vM(x|m_{i-1}, c_{i-1}) \tag{38}$$

$$\propto \exp\{\alpha \cos(x - y) + c_{i-1} \cos(x - m_{i-1})\} \tag{39}$$

The last expression has the form of a von Mises distribution in $x$ and hence:

$$p(x|y; \alpha, m_{i-1}, c_{i-1}) = vM(x; m_i, c_i) \tag{40}$$

where

$$m_i = \text{atan2}(\alpha \sin y + c_{i-1} \sin m_{i-1}, \alpha \cos y + c_{i-1} \cos m_{i-1}) \tag{41}$$

$$c_i = \sqrt{\alpha^2 + c_{i-1}^2 + 2\alpha c_{i-1} \cos(y - m_{i-1})} \tag{42}$$

We refer readers interested in more detailed deduction to Mardia & El-Atoum (1976); Guttorp & Lockhart (1988). Defining the notation for scaler $x$ in circular space as $\dot{\boldsymbol{x}} \overset{\text{def}}{=} [\cos x, \sin x]^T$, these two expressions will be much simpler, intuitive and more similar to the Gaussian form:

$$h(\{\dot{\boldsymbol{m}}_{i-1}, c\}, \dot{\boldsymbol{y}}, \alpha) = \{\dot{\boldsymbol{m}}_i, c_i\} \tag{43}$$

where

$$\dot{\boldsymbol{m}}_i = \frac{\alpha \dot{\boldsymbol{y}} + c_{i-1} \dot{\boldsymbol{m}}_{i-1}}{c_i} \tag{44}$$

$$c_i = ||\alpha \dot{\boldsymbol{y}} + c_{i-1} \dot{\boldsymbol{m}}_{i-1}||_2 \tag{45}$$

The Bayesian update distribution $p_{_U}(\cdot \mid \boldsymbol{\theta}, \mathbf{x}; \alpha)$ could be obtained by marginalizing $\mathbf{y}$:

$$p_U(\theta'|\theta, \mathbf{x}; \alpha) = \mathbb{E}_{p_S(\mathbf{y}|\mathbf{x};\alpha)}\delta(\theta' - h(\theta, \mathbf{y}, \alpha)) = \mathbb{E}_{vM(\mathbf{y}|\mathbf{x},\alpha)}\delta(\theta' - h(\theta, \mathbf{y}, \alpha)) \tag{46}$$

### A.4  *Non-Additive* ACCURACY ISSUE

Although all cases including continuous and discrete data are proven to enjoy the so-called additive accuracy property considered in Graves et al. (2023) defined as:

$$p_{_U}(\boldsymbol{\theta}'' \mid \boldsymbol{\theta}, \mathbf{x}; \alpha_a + \alpha_b) = \mathbb{E}_{p_{_U}(\boldsymbol{\theta}'|\boldsymbol{\theta},\mathbf{x};\alpha_a)}p_{_U}(\boldsymbol{\theta}'' \mid \boldsymbol{\theta}', \mathbf{x}; \alpha_b). \tag{47}$$

this property does not hold for von Mises distribution. The untenability of this property for von Mises distribution can be checked out by considering two steps Bayesian updates with prior $\boldsymbol{\theta} = \{0, 0\}$, $\alpha_a, \alpha_b, \mathbf{y}_a, \mathbf{y}_b$,

$$p_{_U}(c'' \mid \boldsymbol{\theta}, \mathbf{x}; \alpha_a + \alpha_b) = \delta(c - \alpha_a - \alpha_b) \tag{48}$$

$$\neq \mathbb{E}_{p_{_U}(\boldsymbol{\theta}'|\boldsymbol{\theta},\mathbf{x};\alpha_a)}p_{_U}(c'' \mid \boldsymbol{\theta}', \mathbf{x}; \alpha_b) = \mathbb{E}_{vM(\mathbf{y}|\mathbf{x},\alpha_a)}\mathbb{E}_{vM(\mathbf{y}|\mathbf{x},\alpha_b)}\delta(c - ||\alpha_a \mathbf{y}_a + \alpha_b \mathbf{y}_b||) \tag{49}$$

Consequently, the Bayesian flow distribution does not equal to one-step Bayesian update distribution with $\beta(t)$:

$$p_{_F}(\boldsymbol{\theta} \mid \mathbf{x}; t) \neq p_{_U}(\boldsymbol{\theta} \mid \boldsymbol{\theta}_0, \mathbf{x}; \beta(t)). \tag{50}$$

With an accuracy schedule $\alpha_1, \alpha_2, \ldots, \alpha_n$ and $\beta(t) = \sum_{j=1}^{t_i} \alpha_j$, this untenability will cause the incongruity between sender's accumulated accuracy $\beta(t)$ and the confidence of receiver's belief $c_i$ over his location parameter $m_i$. Hence we should differentiate the sender's accuracy schedule $\alpha_i$ and the receiver's belief confidence $c_i$. And $c_i$ is no longer a function but a *distribution* over $t_i$. In consequence, we should define the Bayesian flow distribution parameterized by received sender's accuracies $\alpha_1, \alpha_2, \ldots, \alpha_i$ rather than $t$. Furthermore, the information of receiver confidence $c_i$ should be part of the network input as well.

## A.5 Bayesian Flow Distribution $p_F(\boldsymbol{\theta} \,|\, \mathbf{x}; \alpha_1, \alpha_2, \ldots, \alpha_i)$ and Sender Accuracy Schedule $\alpha_i$

$$p_F(\boldsymbol{\theta} \,|\, \mathbf{x}; \alpha_1, \alpha_2, \ldots, \alpha_i) = \mathbb{E}_{p_U(\boldsymbol{\theta}_1 | \boldsymbol{\theta}_0, \mathbf{x}; \alpha_1)} \ldots \mathbb{E}_{p_U(\boldsymbol{\theta} | \boldsymbol{\theta}_{i-1}, \mathbf{x}; \alpha_i)} p_U(\boldsymbol{\theta} | \boldsymbol{\theta}_{i-1}, \mathbf{x}; \alpha_i) \tag{51}$$

The original definition of Bayesian flow distribution in Eq. (51) provides an iterative algorithm to sample from $p_F$ but which practically is slow resulting the training unaffordable. In fact, noticing the "additive" property of $c_i \dot{\boldsymbol{m}}_i$ by Eqs. (44) and (45), we can sample from $p_F$ without iteration:

$$p_F(\boldsymbol{m} \,|\, \mathbf{x}; \alpha_1, \alpha_2, \ldots, \alpha_i) = \mathbb{E}_{vM(\mathbf{y}_1 \,|\, \mathbf{x}, \alpha_1)} \ldots \mathbb{E}_{vM(\mathbf{y}_i \,|\, \mathbf{x}, \alpha_i)} \delta(\boldsymbol{m} - \operatorname{atan2}(\sum_{j=1}^{i} \alpha_i \cos \mathbf{y}_i, \sum_{j=1}^{i} \alpha_i \sin \mathbf{y}_i)) \tag{52}$$

$$p_F(\boldsymbol{c} \,|\, \mathbf{x}; \alpha_1, \alpha_2, \ldots, \alpha_i) = \mathbb{E}_{vM(\mathbf{y}_1 \,|\, \mathbf{x}, \alpha_1)} \ldots \mathbb{E}_{vM(\mathbf{y}_i \,|\, \mathbf{x}, \alpha_i)} \delta(\boldsymbol{c} - ||(\sum_{j=1}^{i} \alpha_i \cos \mathbf{y}_i, \sum_{j=1}^{i} \alpha_i \sin \mathbf{y}_i)||_2) \tag{53}$$

Eqs. (52) and (53) provides an algorithm allowing sampling from $p_F$ by pure tensor operations without simulating the flow iteratively. Next, we can define the entropy of the receiver's belief as $H(t)$:

$$H(t) \stackrel{\text{def}}{=} \mathbb{E}_{p_F(\boldsymbol{\theta} \,|\, \mathbf{x}; \alpha_1, \alpha_2, \ldots, \alpha_i)} H(p_I(\,\cdot\,|\boldsymbol{\theta})) \tag{54}$$

$$= \mathbb{E}_{p_F(c_i \,|\, \mathbf{x}; \alpha_1, \alpha_2, \ldots, \alpha_i)} - c_i \frac{I_1(c_i)}{I_0(c_i)} + \ln[2\pi I_0(c_i)], \text{ where } i = nt \tag{55}$$

To ensure the information coherence between modalities, we choose to find a sender accuracy schedule to make the receiver's belief entropy $H(t)$ linearly decrease with predefined $c_n$. Formally, we would like to find an accuracy schedule $\alpha_i$ such that

$$H(t) = (1 - t)H(0) + tH(1) = \ln 2\pi - c_n \frac{I_1(c_n)}{I_0(c_n)} + \ln[2\pi I_0(c_n)] \tag{56}$$

Note that Eq. (56) can not be solved analytically but we can solve it numerically by firstly getting the target sender's accumulated accuracy $\beta(t)$ via binary search due to the monotonicity of Eq. (30). Next, we could iteratively search $\alpha_i$ from $i = 1$ to $i = n$ by matching the average accuracy toward $c_i$. This process could be done only once and the resultant $\alpha_i$ can be cached for each pre-confirmed hyper-parameter $(c_n, n)$ pair.

## A.6 Output Distribution $p_O(\cdot \,|\, \boldsymbol{\theta}; t)$ and Receiver Distribution $p_R(\cdot \,|\, \boldsymbol{\theta}; \alpha, t)$

Given samples $\boldsymbol{\theta} = \{\boldsymbol{m}, \boldsymbol{c}\}$ from Bayesian flow distribution as input, the receiver uses network output $\Psi(\boldsymbol{\theta}, t)$ to rebuild his belief over ground truth $\boldsymbol{x}$ termed output distribution. Following Graves et al. (2023), we parameterize $p_O$ using $\hat{\mathbf{x}}(\boldsymbol{\theta}, t) = \Psi(\boldsymbol{\theta}, t)$ to be $\delta$ prediction of $\mathbf{x}$:

$$p_O(\mathbf{x}|\boldsymbol{\theta}; t) = \delta(\mathbf{x} - \hat{\mathbf{x}}(\boldsymbol{\theta}, t)) \tag{57}$$

Therefore, the receiver distribution is:

$$p_R(\mathbf{y}|\boldsymbol{\theta}; \alpha, t) = \mathbb{E}_{p_O(\mathbf{x}'|\theta; t)} p_S(\mathbf{y}|\mathbf{x}'; \alpha) = vM(\mathbf{y}|\hat{\mathbf{x}}(\boldsymbol{\theta}, t), \alpha) \tag{58}$$

## A.7 Discrete-Time Loss $L^n(\mathbf{x})$

From Kitagawa & Rowley (2022), the KL divergence between $vM(m_1, c_1)$ and $vM(m_2, c_2)$ is

$$D_{KL}(vM(m_1, c_1)||vM(m_2, c_2)) = -\ln \frac{I_0(c_1)}{I_0(c_2)} + \frac{I_1(c_1)}{I_0(c_1)}(c_1 \dot{m}_1 - c_2 \dot{m}_2)' \dot{m}_1 \tag{59}$$

From Eq. (5), the discrete-time loss for circular data is

$$L^n(\mathbf{x}) = n\mathbb{E}_{i \sim U\{1, n\}, p_F(\theta | \mathbf{x}; \alpha_1, \alpha_2, \ldots, \alpha_i)} D_{KL}\left(p_S\left(\cdot \,|\, \mathbf{x}; \alpha_i\right) \| p_R(\cdot \,|\, \boldsymbol{\theta}; t_{i-1}, \alpha_i)\right) \tag{60}$$

$$= n\mathbb{E}_{i \sim U\{1, n\}, p_F(\theta | \mathbf{x}; \alpha_1, \alpha_2, \ldots, \alpha_i)} \frac{I_1(\alpha_i)}{I_0(\alpha_i)} \alpha_i (1 - \cos(\mathbf{x} - \hat{\mathbf{x}}(\theta_{i-1}, t_{i-1}))) \tag{61}$$

Continuous-time loss is not tractable because the Bayesian flow distribution is not analytical due to the non-additive accuracy property.

# B  PROOF OF PROPOSITIONS

In this section, we first prove the crystal geometric invariance of CrysBFN. Crystals remain the same under transformations including permutation, orthogonal transformation, and periodic translation defined as follows:

**Definition 1** (Permutation Invariance (Jiao et al., 2023)). *For any permutation matrix $\boldsymbol{P}$, $p(\boldsymbol{L}, \boldsymbol{F}, \boldsymbol{A}) = p(\boldsymbol{L}, \boldsymbol{F}\boldsymbol{P}, \boldsymbol{A}\boldsymbol{P})$,* i.e., *changing the order of atoms will not change the distribution.*

**Definition 2** (O(3) Invariance (Jiao et al., 2023)). *For any orthogonal transformation $\boldsymbol{Q} \in \mathbb{R}^{3 \times 3}$ satisfying $\boldsymbol{Q}^\top \boldsymbol{Q} = \boldsymbol{I}$, $p(\boldsymbol{Q}\boldsymbol{L}, \boldsymbol{F}, \boldsymbol{A}) = p(\boldsymbol{L}, \boldsymbol{F}, \boldsymbol{A})$, namely, any rotation/reflection of $\boldsymbol{L}$ keeps the distribution unchanged.*

**Definition 3** (Periodic Translation Invariance (Jiao et al., 2023)). *For any translation $\boldsymbol{t} \in \mathbb{R}^{3 \times 1}$, $p(\boldsymbol{L}, w(\boldsymbol{F} + \boldsymbol{t}\boldsymbol{1}^\top), \boldsymbol{A}) = p(\boldsymbol{L}, \boldsymbol{F}, \boldsymbol{A})$, where the function $w(\boldsymbol{F}) = \boldsymbol{F} - \lfloor \boldsymbol{F} \rfloor \in [0, 1)^{3 \times N}$ returns the fractional part of each element in $\boldsymbol{F}$, and $\boldsymbol{1} \in \mathbb{R}^{3 \times 1}$ is a vector with all elements set to one. It explains that any periodic translation of $\boldsymbol{F}$ will not change the distribution.*

The combination of the above invariances is abbreviated in a compact manner termed *periodic E(3) invariance* proposed by Jiao et al. (2023). The permutational invariance can be easily achieved by using GNN frameworks. Periodic translation and rotation are both space group transformations. We first introduce the basic concept of *G-invariant*.

**Definition 4.** *We call a distribution $p(x)$ is G-invariant if for any transformation $g$ in the group $G$, $p(g \cdot x) = p(x)$, and a conditional distribution $p(x|c)$ is G-equivariant if $p(g \cdot x|g \cdot c) = p(x|c), \forall g \in G$.*

With a lemma from Xu et al. (2021), we can prove a Markov-process-generated distribution *G-invariant* by proving the G-invariance of the prior distribution and the G-equivariance of every transition kernel.

**Lemma 1** (Xu et al. (2021)). *Consider the generation Markov process $p(\theta_n) = \int p(\theta_0)p(\theta_{n:1}|\theta_0)d\theta_{1:n}$. If the prior distribution $p(\theta_0)$ is G-invariant and the Markov transitions $p(\theta_{t+1}|\theta_t), 0 \le t \le n-1$ are G-equivariant, the marginal distribution $p(\theta_n)$ is also G-invariant.*

*Proof.*

$$\forall g \in G, \quad p(g \cdot \theta_n) = p(g \cdot \theta_0) \int p(\theta_{n:1}|\theta_0)d\theta_{1:n}$$

$$= p(g \cdot \theta_0) \int \prod_{t=0}^{n-1} p(g \cdot \theta_{t+1}|g \cdot \theta_t)d\theta_{1:n}$$

$$= p(\theta_0) \int \prod_{t=0}^{n-1} p(g \cdot \theta_{t+1}|g \cdot \theta_t)d\theta_{1:n}$$

$$= p(\theta_0) \int \prod_{t=0}^{n-1} p(\theta_{t+1}|\theta_t)d\theta_{1:n}$$

$$= p(\theta_0) \int p(\theta_0)p(\theta_{n:1}|\theta_0)d\theta_{1:n}$$

$$= p(\theta_n).$$

Therefore, the marginal distribution $p(\theta_n)$ is G-invariant. $\square$

With Lemma 1, we can prove the following propositions mentioned in the main text:

**Proposition 4.3.** *With $\Psi_L$ as O(3)-equivariant function namely $\Psi_L(\boldsymbol{\theta}^A, \boldsymbol{\theta}^F, \boldsymbol{Q}\boldsymbol{\theta}^L, t) = \boldsymbol{Q}\Psi_L(\boldsymbol{\theta}^A, \boldsymbol{\theta}^F, \boldsymbol{\theta}^L, t), \forall \boldsymbol{Q}^T \boldsymbol{Q} = \boldsymbol{I}$, the marginal distribution of $p(\boldsymbol{\mu}_n^L)$ defined by Eq. (22) is O(3)-invariant.*

*Proof.* The prior is O(3) invariant since $p(\boldsymbol{\mu}_0^L) = \delta(\boldsymbol{\mu} - \mathbf{0}) = \delta(\boldsymbol{Q}\boldsymbol{\mu} - \mathbf{0}), \forall \boldsymbol{Q}^T\boldsymbol{Q} = \boldsymbol{I}$. The transition probability of

$$
\begin{aligned}
&p(\boldsymbol{Q}\boldsymbol{\mu}_i^L | \boldsymbol{Q}\boldsymbol{\mu}_{i-1}^L, \boldsymbol{\theta}_{i-1}^A, \boldsymbol{\theta}_{i-1}^F) \\
=&p_U^L(\boldsymbol{Q}\boldsymbol{\mu}_i^L | \boldsymbol{Q}\boldsymbol{\mu}_{i-1}^L, \hat{\Psi}_L(\boldsymbol{Q}\boldsymbol{\mu}_{i-1}^L, \cdot), t_{i-1}) \\
=&\mathcal{N}(\boldsymbol{Q}\boldsymbol{\mu}_i^L | \frac{\alpha\hat{\Psi}_L(\boldsymbol{Q}\boldsymbol{\mu}_{i-1}^L, \cdot) + \boldsymbol{Q}\boldsymbol{\mu}_{i-1}^L\rho_{i-1}}{\rho_i}, \frac{\alpha}{\rho_i^2}\boldsymbol{I}) \\
=&\mathcal{N}(\boldsymbol{Q}\boldsymbol{\mu}_i^L | \frac{\alpha\boldsymbol{Q}\hat{\Psi}_L(\boldsymbol{\mu}_{i-1}^L, \cdot) + \boldsymbol{Q}\boldsymbol{\mu}_{i-1}^L\rho_{i-1}}{\rho_i}, \frac{\alpha}{\rho_i^2}\boldsymbol{I}) \quad \text{(by equivariance of } \Psi^L) \\
=&\mathcal{N}(\boldsymbol{Q}\boldsymbol{\mu}_i^L | \boldsymbol{Q}\frac{\alpha\hat{\Psi}_L(\boldsymbol{\mu}_{i-1}^L, \cdot) + \boldsymbol{\mu}_{i-1}^L\rho_{i-1}}{\rho_i}, \frac{\alpha}{\rho_i^2}\boldsymbol{I}) \\
=&\mathcal{N}(\boldsymbol{\mu}_i^L | \frac{\alpha\hat{\Psi}_L(\boldsymbol{\mu}_{i-1}^L, \cdot) + \boldsymbol{\mu}_{i-1}^L\rho_{i-1}}{\rho_i}, \frac{\alpha}{\rho_i^2}\boldsymbol{I}) \quad \text{(by property of isotropic Normal p.d.f)} \\
=&p_U^L(\boldsymbol{\mu}_i^L | \boldsymbol{\mu}_{i-1}^L, \hat{\Psi}_L(\boldsymbol{\mu}_{i-1}^L, \cdot), t_{i-1}) \\
=&p(\boldsymbol{\mu}_i^L | \boldsymbol{\mu}_{i-1}^L, \boldsymbol{\theta}_{i-1}^A, \boldsymbol{\theta}_{i-1}^F)
\end{aligned}
$$

is equivariant. By Lemma 1, the marginal distribution $p(\boldsymbol{\mu}_n^L)$ is O(3)-invariant. $\square$

**Proposition 4.2.** *With $\Psi_F$ as a periodic translation equivariant function namely $\Psi_F(\boldsymbol{\theta}^A, w(\boldsymbol{\theta}^F + \boldsymbol{t}), \boldsymbol{\theta}^L, t) = w(\Psi_F(\boldsymbol{\theta}^A, \boldsymbol{\theta}^F, \boldsymbol{\theta}^L, t) + \boldsymbol{t}), \forall \boldsymbol{t} \in \mathbb{R}^3$, the marginal distribution of $p(\boldsymbol{F}_n)$ defined by Eqs. (17) and (19) is periodic translation invariant.*

*Proof.* We first prove that the Bayesian update is periodic translational equivariant. Based on Eqs. (44) and (45), we can interpret the Bayesian update of $\{\dot{\boldsymbol{m}}_{i-1}, c_{i-1}\}$ observing $\dot{\boldsymbol{y}}$ with $\alpha$, as the *vector addition* between $\dot{\boldsymbol{m}}_{i-1}$ and $\dot{\boldsymbol{y}}$ with weight $c_{i-1}$ and $\alpha$. And the periodic translation $t$ for $x$ corresponds to the rotation of $\dot{\boldsymbol{x}}$ with angle $t$:

$$
\begin{aligned}
&[\cos(x + t + 2\pi k), \sin(x + t + 2\pi k)]^T = [\cos(x + t), \sin(x + t)]^T \\
=&[\cos x \cos t - \sin x \sin t, \sin x \cos t + \cos x \sin t]^T \\
=&\begin{bmatrix} \cos t & -\sin t \\ \sin t & \cos t \end{bmatrix} \begin{bmatrix} \cos x \\ \sin x \end{bmatrix} = \boldsymbol{R}_t\dot{\boldsymbol{x}}
\end{aligned}
$$

where $\boldsymbol{R}_t$ is the 2-dimensional rotation matrix with angle $t$. Due to the rotational equivariance of 2D vector addition, we can infer that the Bayesian update function $h$ is periodic translational equivariant:

$$
\begin{aligned}
&h(\{w(m_{i-1} + t), c_{i-1}\}, w(y + t), \alpha) = h(\{\boldsymbol{R}_t\dot{\boldsymbol{m}}_{i-1}, c_{i-1}\}, \boldsymbol{R}_t\dot{\boldsymbol{y}}, \alpha) \\
=&\{\frac{\alpha\boldsymbol{R}_t\dot{\boldsymbol{y}} + c_{i-1}\boldsymbol{R}_t\dot{\boldsymbol{m}}_{i-1}}{||\alpha\boldsymbol{R}_t\dot{\boldsymbol{y}} + c_{i-1}\boldsymbol{R}_t\dot{\boldsymbol{m}}_{i-1}||_2}, ||\alpha\boldsymbol{R}_t\dot{\boldsymbol{y}} + c_{i-1}\boldsymbol{R}_t\dot{\boldsymbol{m}}_{i-1}||_2\} \\
=&\{\frac{\alpha\boldsymbol{R}_t\dot{\boldsymbol{y}} + c_{i-1}\boldsymbol{R}_t\dot{\boldsymbol{m}}_{i-1}}{||\alpha\dot{\boldsymbol{y}} + c_{i-1}\dot{\boldsymbol{m}}_{i-1}||_2}, ||\alpha\dot{\boldsymbol{y}} + c_{i-1}\dot{\boldsymbol{m}}_{i-1}||_2\} \\
=&\{\boldsymbol{R}_t\dot{\boldsymbol{m}}_i, c_i\} = \{w(m_i + t), c_i\}
\end{aligned}
\tag{62}
$$

The prior is periodic translation invariant because $\boldsymbol{m}^F \sim U(\mathbf{0}, \mathbf{1})$.
We prove that the Bayesian update distribution $p_U(\boldsymbol{m}_i^F | \boldsymbol{m}_{i-1}^F, \boldsymbol{c}_{i-1}^F, g(\boldsymbol{m}_{i-1}^F); \alpha)$ is periodic transla-

tion equivariant if $\Psi^F$ is periodic translation equivariant:

$$p_U(w(\boldsymbol{m}_i^F + \boldsymbol{t})|w(\boldsymbol{m}_{i-1}^F + \boldsymbol{t}), \boldsymbol{c}_{i-1}^F, \Psi^F(w(\boldsymbol{m}_{i-1}^F + \boldsymbol{t})); \alpha)$$

$$=\mathbb{E}_{vM(\mathbf{y}|\Psi^F(w(\boldsymbol{m}_{i-1}^F+\boldsymbol{t})),\alpha)}\delta(w(\boldsymbol{m}_i^F + \boldsymbol{t}) - h(w(\boldsymbol{m}_{i-1}^F + \boldsymbol{t}), \boldsymbol{c}_{i-1}^F, \mathbf{y}, \alpha))$$

$$=\mathbb{E}_{vM(\mathbf{y}|w(\Psi^F(\boldsymbol{m}_{i-1}^F)+\boldsymbol{t}),\alpha)}\delta(w(\boldsymbol{m}_i^F + \boldsymbol{t}) - h(w(\boldsymbol{m}_{i-1}^F + \boldsymbol{t}), \boldsymbol{c}_{i-1}^F, \mathbf{y}, \alpha))(\text{by equiv.}\Psi^F)$$

$$=\mathbb{E}_{vM(w(\mathbf{y}+\boldsymbol{t})|w(\Psi^F(\boldsymbol{m}_{i-1}^F)+\boldsymbol{t}),\alpha)}\delta(w(\boldsymbol{m}_i^F + \boldsymbol{t}) - h(w(\boldsymbol{m}_{i-1}^F + \boldsymbol{t}), \boldsymbol{c}_{i-1}^F, w(\mathbf{y}+\boldsymbol{t}), \alpha))$$

$$=\mathbb{E}_{vM(\mathbf{y}|\Psi^F(\boldsymbol{m}_{i-1}^F)),\alpha)}\delta(w(\boldsymbol{m}_i^F + \boldsymbol{t}) - h(w(\boldsymbol{m}_{i-1}^F + \boldsymbol{t}), \boldsymbol{c}_{i-1}^F, w(\mathbf{y}+\boldsymbol{t}), \alpha))(\text{by Eq. (29)})$$

$$=\mathbb{E}_{vM(\mathbf{y}|\Psi^F(\boldsymbol{m}_{i-1}^F)),\alpha)}\delta(w(\boldsymbol{m}_i^F + \boldsymbol{t}) - w(h(\boldsymbol{m}_{i-1}^F, \boldsymbol{c}_{i-1}^F, \mathbf{y}, \alpha) + \boldsymbol{t}))(\text{by Eq. (62)})$$

$$=\mathbb{E}_{vM(\mathbf{y}|\Psi^F(\boldsymbol{m}_{i-1}^F)),\alpha)}\delta(\boldsymbol{m}_i^F - h(\boldsymbol{m}_{i-1}^F, \boldsymbol{c}_{i-1}^F, \mathbf{y}, \alpha))(\text{by equivariance of }\delta\text{ function})$$

$$=p_U(\boldsymbol{m}_i^F|\boldsymbol{m}_{i-1}^F, \boldsymbol{c}_{i-1}^F, \Psi^F(\boldsymbol{m}_{i-1}^F); \alpha)$$

$\square$

Next, we prove the following proposition:

**Proposition 4.1.** *The probability density function of Bayesian flow distribution defined by Eqs. (15) and (16) is equivalent to the original definition in Eq. (14).*

*Proof.* Combining Eqs. (44) and (45),

$$\dot{\boldsymbol{m}}_i\boldsymbol{c}_i = \alpha_i\dot{\mathbf{y}}_i + \boldsymbol{c}_{i-1}\dot{\boldsymbol{m}}_{i-1}$$
$$= \alpha_i\dot{\mathbf{y}}_i + \alpha_{i-1}\dot{\mathbf{y}}_{i-1} + \boldsymbol{c}_{i-2}\dot{\boldsymbol{m}}_{i-2}$$
$$= \alpha_i\dot{\mathbf{y}}_i + \cdots + \alpha_1\dot{\mathbf{y}}_1 + \boldsymbol{c}_0\dot{\boldsymbol{m}}_0$$
$$= \sum_{j=1}^{i}\alpha_i\dot{\mathbf{y}}_i = [\sum_{j=1}^{i}\alpha_i\cos\mathbf{y}_i, \sum_{j=1}^{i}\alpha_i\sin\mathbf{y}_i]^T$$

Taking the 2-norm to each side,

$$||\dot{\boldsymbol{m}}_i\boldsymbol{c}_i||_2 = ||[\sum_{j=1}^{i}\alpha_i\cos\mathbf{y}_i, \sum_{j=1}^{i}\alpha_i\sin\mathbf{y}_i]^T||_2$$

$$\boldsymbol{c}_i = ||[\sum_{j=1}^{i}\alpha_i\cos\mathbf{y}_i, \sum_{j=1}^{i}\alpha_i\sin\mathbf{y}_i]^T||_2$$

The vector direction of $\dot{\boldsymbol{m}}_i$ is irrelevant to the scaler $\boldsymbol{c}_i$. Therefore,

$$\dot{\boldsymbol{m}}_i = \text{atan2}(\sum_{j=1}^{i}\alpha_i\cos\mathbf{y}_i, \sum_{j=1}^{i}\alpha_i\sin\mathbf{y}_i) \tag{63}$$

Hence,

$$p_F(\boldsymbol{m}_i|\mathbf{x}; \alpha_1, \alpha_2, \ldots, \alpha_i)$$

$$=\mathbb{E}_{p_U(\boldsymbol{\theta}_1|\boldsymbol{\theta}_0,\mathbf{x};\alpha_1)}\cdots\mathbb{E}_{p_U(\boldsymbol{\theta}_{i-1}|\boldsymbol{\theta}_{i-2},\mathbf{x};\alpha_{i-1})}p_U(\boldsymbol{m}_i|\boldsymbol{\theta}_{i-1}, \mathbf{x}; \alpha_i)$$

$$=\mathbb{E}_{vM(\mathbf{y}_1|\mathbf{x},\alpha_1)}\cdots\mathbb{E}_{vM(\mathbf{y}_i|\mathbf{x},\alpha_i)}\delta(\boldsymbol{m}_i - \text{atan2}(\sum_{j=1}^{i}\alpha_i\cos\mathbf{y}_i, \sum_{j=1}^{i}\alpha_i\sin\mathbf{y}_i))$$

$$p_F(\boldsymbol{c}_i|\mathbf{x}; \alpha_1, \alpha_2, \ldots, \alpha_i)$$

$$=\mathbb{E}_{p_U(\boldsymbol{\theta}_1|\boldsymbol{\theta}_0,\mathbf{x};\alpha_1)}\cdots\mathbb{E}_{p_U(\boldsymbol{\theta}_{i-1}|\boldsymbol{\theta}_{i-2},\mathbf{x};\alpha_{i-1})}p_U(\boldsymbol{m}_i|\boldsymbol{\theta}_{i-1}, \mathbf{x}; \alpha_i)$$

$$=\mathbb{E}_{vM(\mathbf{y}_1|\mathbf{x},\alpha_1)}\cdots\mathbb{E}_{vM(\mathbf{y}_i|\mathbf{x},\alpha_i)}\delta(\boldsymbol{c}_i - ||[\sum_{j=1}^{i}\alpha_i\cos\mathbf{y}_i, \sum_{j=1}^{i}\alpha_i\sin\mathbf{y}_i]^T||_2)$$

$\square$

## C  IMPLEMENTATION DETAILS

**Training and Sampling Procedure** We provide the training and sampling procedure in Algorithm 1 and in Algorithm 2.

**Network Architecture** We use CSPNet proposed by Jiao et al. (2023) with minor modifications: (1) We add a residual connection from the input to the output of fractional coordinates ensuring the equivariance of the network:

$$\Psi^F(\boldsymbol{\theta}_i^A, \boldsymbol{\theta}_i^F, \boldsymbol{\theta}_i^L, t_i) = w(\varphi_F(\boldsymbol{h}_i^{(S)}) + \boldsymbol{\theta}_i^F), \tag{64}$$

By the periodic translational invariance of $\varphi_F(\boldsymbol{h}_i^{(S)})$ proved by Jiao et al. (2023), the equivariance of $\hat{\Psi}_F(\boldsymbol{\theta}_i^{\mathcal{M}}, t_i)$ can be easily checked:

$$\begin{aligned}
\Psi^F(\boldsymbol{\theta}_i^A, w(\boldsymbol{\theta}_i^F + \boldsymbol{t}), \boldsymbol{\theta}_i^L, t_i) &= w(\varphi_F(\boldsymbol{h}_i^{(S)}) + w(\boldsymbol{\theta}_i^F + \boldsymbol{t})) \\
&= w(w(\varphi_F(\boldsymbol{h}_i^{(S)}) + \boldsymbol{\theta}_i^F) + \boldsymbol{t}) \\
&= w(\Psi^F(\boldsymbol{\theta}_i^A, \boldsymbol{\theta}_i^F, \boldsymbol{\theta}_i^L, t_i) + \boldsymbol{t})
\end{aligned}$$

(2) We alter the frequency of the Fourier transformation features to model in the interval $[-\pi, \pi)$ with length $2\pi$. (3) The concentration parameter of each fractional coordinate $\boldsymbol{c}^F$ is taken logarithm, normalized and concatenated to time embedding. The network hyper-parameters follows the setting of Jiao et al. (2023) including the number of hidden states and layers.

**Hyper-parameters** For the network, the CSPNet has 6 layers, 512 hidden states, 128 frequencies for the Fourier feature for each task and dataset following (Jiao et al., 2023). For BFN hyper-parameters, we set $\sigma_1^2 = 0.001$ for continuous variable generation, $\beta_1 = 1000$ for circular variables generation across all datasets and tasks. For discrete variables, we set $\beta_1 = 0.4$ for the MP-20 dataset and $\beta_1 = 3.0$ for the Perov-5 dataset. The number of steps is searched in $\{50, 100, 500, 1000, 2000\}$. For optimizations, we apply an AdamW optimizer with an initial learning rate $1 \times 10^{-3}$ and a plateau scheduler with a decaying factor of 0.6, a patience of 100 epochs, and a minimal learning rate $1 \times 10^{-4}$. The weight of every loss is $5 \times 10^{-2}$. The network is trained for 4000, 5000, 1500, and 1000 epochs for Perov-5, Carbon-24, MP-20, and MPTS-52 respectively.

**Computational Resources** All training experiments are conducted on a server with $8 \times$ NVidia RTX 3090 GPU, $64 \times$ Intel Xeon Platinum 8362 CPU and 256GB memory. Each training task requires one GPU. We also report the required GPU hour across methods to converge in our experimental environment in Tab. 4.

Table 4: Comparison of GPU hours required for training across different methods.

| GPU Hour | Perov-5 | MP-20 | MPTS-52 |
|---|---|---|---|
| DiffCSP Jiao et al. (2023) | 8.59 | 92.22 | 10.42 |
| FlowMM Miller et al. (2024) | 16.36 | 106.37 | 16.49 |
| CrysBFN | 10.19 | 85.71 | 12.31 |

## D  MORE RESULTS

**Visualizations** Here we give visualizations of the ab-initio generated structures from CrysBFN and DiffCSP in Fig. 6. We also provide a gif animation of the generation process in our code repository `https://github.com/wu-han-lin/CrysBFN`.

**Error Bars** We report the error bar for the crystal structure prediction task following Jiao et al. (2023) in Tab. 5 running three experiments with different random seeds. The results are similar to Tab. 2.

**Uniqueness, Novelty, and Stability** Here we compare the uniqueness, novelty, and stability of ab initio generated samples across methods on MP-20. Using StructureMatcher in pymatgen with default parameters, a generated crystal is considered: 1) unique if it does not match any other generated samples; and 2) novel if it does not match any crystals in the training set, following prior practices (Zeni et al., 2023; Miller et al., 2024). The stability evaluation procedure follows the approach in Gruver et al. (2024). Finally, a sample is considered stable, unique, and novel (S.U.N.) if it satisfies all three conditions. The results are reported in Tab. 6.

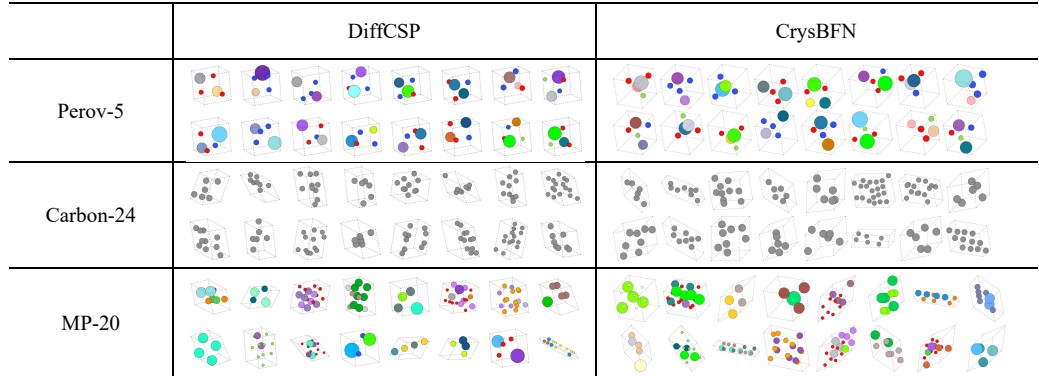

|  | DiffCSP | CrysBFN |
|---|---|---|
| Perov-5 |  |  |
| Carbon-24 |  |  |
| MP-20 |  |  |

Figure 6: Visualizations comparison of the ab-initio generated structures from CrysBFN and DiffCSP.

Table 5: Results on Perov-5 and MP-20 with error bars.

|  | Perov-5 | | MP-20 | |
|---|---|---|---|---|
|  | Match rate (%)↑ | RMSE↓ | Match rate (%)↑ | RMSE↓ |
| CDVAE (Xie et al., 2021) | 45.31±0.49 | 0.1123±0.0026 | 33.93±0.15 | 0.1069±0.0018 |
| DiffCSP (Jiao et al., 2023) | 52.35±0.26 | 0.0778±0.0030 | 51.89±0.30 | 0.0611±0.0015 |
| CrysBFN | **54.58±0.13** | **0.0691±0.0011** | **64.33±0.24** | **0.0445±0.0010** |

# E    SAMPLING EFFICIENCY COMPARISON TO ODE SAMPLERS

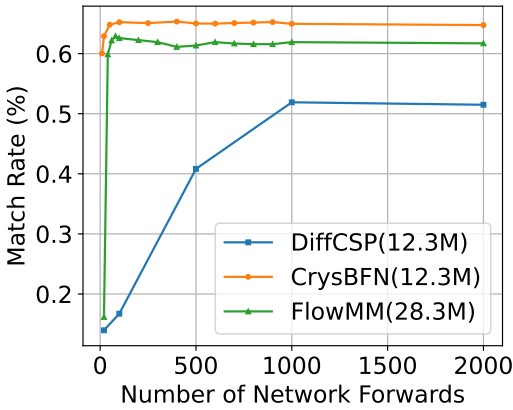

Figure 7: Experimental results on MP-20 with different Number of Function Evaluations (NFE) *i.e.* number of network forward passes including FlowMM. Note that FlowMM has a larger parameter size resulting in a less fair comparison.

We present the comparison among them in Fig. 7 and find that FlowMM fails in extremely small NFE (20 steps) settings with only 16.18% match rate, while CrysBFN enjoys 60.02% match rate with 10 steps and consistently achieves the best sampling quality.

# F    DETAILED DISCUSSION OF RELATED WORKS

Discovering new functional materials has been a long-standing scientific problem. Recently, data-driven approaches have been seen as a promising solution to address this challenge  Peng et al. (2022).

**Two-stage crystal generation methods based on implicit crystal representations** One line of approaches indirectly generates crystals in the implicit representation space.  Prior practices in-

Table 6: Uniqueness, novelty, and stability experimental results comparison on ab initio generation task on MP-20 dataset.

| Method | Unique / % | Novel / % | Metastable / % | Stable / % | S.U.N. Rate / % |
|---|---|---|---|---|---|
| DiffCSP (Jiao et al., 2023) | **96.11** | 90.95 | 37.91 | 12.16 | 9.44 |
| FlowMM (Miller et al., 2024) | 94.79 | 91.63 | 32.77 | 9.23 | 8.31 |
| CrysBFN | 95.29 | **92.37** | **45.91** | **15.82** | **12.16** |

Table 7: Ablation study results of entropy-conditioning mechanism across datasets.

| | Perov-5 | | MP-20 | | MPTS-52 | |
|---|---|---|---|---|---|---|
| | Match rate↑ | RMSE↓ | Match rate↑ | RMSE↓ | Match rate↑ | RMSE↓ |
| w/o entropy conditioning | 51.33 | 0.0753 | 52.16 | 0.0631 | 13.41 | 0.1547 |
| CrysBFN | **54.69** | **0.0636** | **64.35** | **0.0433** | **20.52** | **0.1038** |

cludes transforming crystals into human-designed fingerprint FTCP (Ren et al., 2021), 3D voxels images (Hoffmann et al., 2019), 2D images (Noh et al., 2019), 3D electron-density maps (Court et al., 2020), video-like representation (Yang et al., 2023), embedded atom density (Zhang et al., 2019) in StructRepDiff (Sinha et al., 2024). However, their generation quality is hampered by the encoding and reconstruction processes, which may not be fully reversible or fail to respect physical symmetries such as rotational and translational invariance. For example, 3D voxel grids (Hoffmann et al., 2019) and 3D density maps (Court et al., 2020) are invariant to periodic transformations but not to $E(3)$ transformations (Zhang et al., 2023), and the video-like representation (Yang et al., 2023) is not invariant to permutation, rotation, translation, and periodic transformations.

**Direct crystal generation methods** Direct material generation in sample space could bypass the above reversibility problem. Prior works (Nouira et al., 2018; Kim et al., 2020) employ Generative Adversarial Networks (Goodfellow et al., 2020) to generate crystal structures, while their methods fail to respect crystal geometric invariance. Inspired by the success of Diffusion models on images (Ho et al., 2020b; Song et al., 2020a; Song & Ermon, 2019), the multi-step generation paradigm has been introduced into generative modeling of atom systems including molecular conformations (Xu et al., 2023). The geometric invariance of the generation path can be guaranteed by designing a Markov chain with an invariant prior and equivariant transitions (Xu et al., 2021). CDVAE (Xie et al., 2021), its CSP adaptation Cond-CDVAE (Luo et al., 2024a) and SyMat (Luo et al., 2024b), generate crystalline materials leveraging $E(3)$-equivariant graph neural network models (Klipfel et al., 2023; Gasteiger et al., 2021a) on 3D multi-edge graphs. Utilizing VAE models, they generate lattice parameters, randomly initialize atom coordinates, and iteratively refine these coordinates using score-matching models Song & Ermon (2019). With the fractional coordinate system, DiffCSP (Jiao et al., 2023) firstly introduces the periodic E(3) equivariance of crystals and designs an equivariant diffusion crystal generation model based on periodic diffusion (Jing et al., 2022). Subsequently, FlowMM (Miller et al., 2024) recently introduced Riemannian Flow Matching (Chen & Lipman, 2023) for the task of crystal generation, offering improved sampling efficiency, albeit at the expense of quality.

We argue that such struggles of balancing between sampling quality and efficiency stems from the lack of proper guidance on each transition from noise prior to data distribution especially for crystals, where thermodynamically stable materials represent only a small fraction in the search space (Miller et al., 2024). For example, early generation states $x_{t-1}$ with low confidence should be preserved less than the later ones to get the next state $x_t$. From the perspective of Bayesian updates, Bayesian Flow Networks (Graves et al., 2023) provides a framework to accurately update each $m_{t-1}$ according to its confidence parameter $\alpha_i$, the effectiveness of which has been proved in Song et al. (2023). However, periodicity is not considered in Graves et al. (2023) and incorporating it into BFN is non-trivial without desirable distributions with mathematical properties like Gaussian. To address above issues, in this paper, we build a Bayesian flow almost from scratch, identifying and tackling the non-additive accuracy via introducing a novel entropy conditioning mechanism, theoretical reformulations of BFN,

a fast sampling algorithm, *etc*. We demonstrate the effectiveness of the guidance of entropy in Tab. 3 and Tab. 7 and its higher sampling effiency and quality in Figs. 4 and 7.

Additionally, recently various techniques have also been introduced to boost the performance tailored for crystal generation considering crystal's inductive bias, including Jiao et al. (2024); Cao et al. (2024); AI4Science et al. (2023) which incorporate the space group constraint into the generation process. Recently, EquiCSP (Lin et al., 2024) proposed to utilize a periodic CoM-free noising method and introduce lattice permutation invariance loss. Those techniques are orthogonal to the proposed method in this paper.

---

**Algorithm 1** Training Procedure

---

1: **Require:** number of steps $n \in \mathbb{R}$, $\sigma_1 \in \mathbb{R}^+$, $\beta_1 \in \mathbb{R}^+$, $\alpha_1^F, \ldots, \alpha_n^F \in \mathbb{R}^+$.
2: **Input:** atom types $\boldsymbol{A}$, fractional coordinates $\boldsymbol{F}$, lattice parameter $\boldsymbol{L}$, length of vocabulary $K$
3: Sample $i \sim U\{1, n\}$, $t \leftarrow \frac{(i-1)}{n}$
4: # sampling from Bayesian flow distribution of lattice
5: $\gamma^L \leftarrow 1 - \sigma_1^{2t}$
6: $\boldsymbol{\mu}_L \sim \mathcal{N}(\gamma^L \boldsymbol{L}, \gamma^L(1 - \gamma^L)\boldsymbol{I})$
7: # sampling from Bayesian flow distribution of atom type
8: $\beta^A \leftarrow \beta_1 t^2$
9: $\mathbf{y}'_A \sim \mathcal{N}(\beta(K\mathbf{e_x} - \mathbf{1}KD), \beta K\boldsymbol{I})$
10: $\boldsymbol{\theta}^A \leftarrow \text{softmax}(\mathbf{y}'_A)$
11: # sampling from Bayesian flow distribution of fractional coordinates
12: $\mathbf{y}_1, \mathbf{y}_2, \ldots, \mathbf{y}_i \leftarrow vM(\boldsymbol{F}, \alpha_1^F), \ldots vM(\boldsymbol{F}, \alpha_i^F)$
13: $\boldsymbol{m}_i \leftarrow \text{atan2}(\sum_{j=1}^{i} \alpha_j \cos \mathbf{y}_j, \sum_{j=1}^{i} \alpha_j \sin \mathbf{y}_j)$
14: # calculate the accumulated accuracy *i.e.* entropy
15: $\boldsymbol{c}_i \leftarrow ||[\sum_{j=1}^{i} \alpha_j \cos \mathbf{y}_j, \sum_{j=1}^{i} \alpha_j \sin \mathbf{y}_j]^T||_2$
16: # use network to do inter-dependency modeling across dimensions conditioning on entropy
17: $\boldsymbol{\theta}^{\mathcal{M}} \leftarrow (\boldsymbol{\mu}_L, \boldsymbol{\theta}^A, \boldsymbol{m}_i, \boldsymbol{c}_i)$
18: $\hat{\Psi}_L(\boldsymbol{\theta}^{\mathcal{M}}_{i-1}, t_{i-1}), \hat{\Psi}_F(\boldsymbol{\theta}^{\mathcal{M}}_{i-1}, t_{i-1}), \hat{\Psi}_F(\boldsymbol{\theta}^{\mathcal{M}}_{i-1}, t_{i-1}) \leftarrow \Psi(\boldsymbol{\theta}^{\mathcal{M}}, t)$
19: # calculate the losses of all modalities
20: $\alpha_i^A \leftarrow \beta_1^A \left(\frac{2i-1}{n^2}\right)$
21: $\mathbf{y}_A \sim \mathcal{N}(\alpha(K\mathbf{e_x} - \mathbf{1}KD), \alpha K\boldsymbol{I})$
22: $\mathcal{L}_A \leftarrow n \ln \mathcal{N}\left(\mathbf{y}_A \mid \alpha_i^A(K\mathbf{e}_{\boldsymbol{A}} - \mathbf{1}), \alpha_i^A K\boldsymbol{I}\right)$
23: $\qquad - \sum_{d=1}^{N} \ln\left(\sum_{k=1}^{K} p_O^{(d)}(k \mid \boldsymbol{\theta}^A; t_{i-1}) \mathcal{N}\left(y_A^{(d)} \mid \alpha_i^A(K\mathbf{e}_k - \mathbf{1}), \alpha_i^A K\boldsymbol{I}\right)\right)$
24: $\mathcal{L}_F \leftarrow n \, \alpha_i^F \frac{I_1(\alpha_i^F)}{I_0(\alpha_i^F)}(1 - \cos(\boldsymbol{F} - \hat{\Psi}_F(\boldsymbol{\theta}^{\mathcal{M}}_{i-1}, t_{i-1})))$
25: $\mathcal{L}_L = \frac{n}{2}\left(1 - \sigma_1^{2/n}\right) \frac{\left\|\boldsymbol{L} - \hat{\Psi}_L(\boldsymbol{\theta}^{\mathcal{M}}_{i-1}, t_{i-1})\right\|^2}{\sigma_1^{2i/n}}$
26: Minimize $\mathcal{L}_A + \mathcal{L}_F + \mathcal{L}_L$

---

---

**Algorithm 2** Sampling Procedure

---

**Require:** number of steps $n \in \mathbb{R}$, length of vocabulary $K$, $\sigma_1 \in \mathbb{R}^+$, $\beta_1 \in \mathbb{R}^+$, $\alpha_1^F, \ldots, \alpha_n^F \in \mathbb{R}^+$

\# initialize the prior parameters

$\boldsymbol{\mu}_0, \rho_0 \leftarrow \mathbf{1}$

$\boldsymbol{\theta}_0 \leftarrow \frac{1}{K}\mathbf{1}_{1 \times N}$

$\boldsymbol{m}_0 \leftarrow U(\mathbf{0}, \mathbf{1}), \boldsymbol{c}_0 \leftarrow \mathbf{0}$

**for** $i \leftarrow 1, \cdots, n$ **do**

    $t \leftarrow \frac{i-1}{n}$

    \# use network to do inter-dependency modeling across dimensions of all modalities

    $\boldsymbol{\theta}^{\mathcal{M}} \leftarrow (\boldsymbol{\mu}_{i-1}, \boldsymbol{\theta}_{i-1}, \boldsymbol{m}_{i-1}, \boldsymbol{c}_{i-1})$

    $\hat{\Psi}_L(\boldsymbol{\theta}_{i-1}^{\mathcal{M}}, t_{i-1}), \hat{\Psi}_F(\boldsymbol{\theta}_{i-1}^{\mathcal{M}}, t_{i-1}), \hat{\Psi}_F(\boldsymbol{\theta}_{i-1}^{\mathcal{M}}, t_{i-1}) \leftarrow \Psi(\boldsymbol{\theta}^{\mathcal{M}}, t)$

    **if** $i < n$ **then**

        \# do Bayesian update for lattice parameter

        $\alpha_i^L \leftarrow \sigma_1^{-2i/n}(1 - \sigma_1^{2/n})$

        $\mathbf{y}^L \sim \mathcal{N}(\hat{\Psi}_L(\boldsymbol{\theta}_{i-1}^{\mathcal{M}}, t_{i-1}), \frac{1}{\alpha_i^L}\boldsymbol{I})$

        $\boldsymbol{\mu}_i \leftarrow \frac{\rho_{i-1}\boldsymbol{\mu}_{i-1} + \alpha_i^L \mathbf{y}^L}{\rho_{i-1} + \alpha_i^L}$

        $\rho_i \leftarrow \rho_{i-1} + \alpha_i^L$

        \# do Bayesian update for fractional coordinates

        $\mathbf{y}^F \sim vM(\hat{\Psi}_F(\boldsymbol{\theta}_{i-1}^{\mathcal{M}}, t_{i-1}), \alpha_i^F)$

        $\boldsymbol{m}_i \leftarrow \text{atan2}(\alpha_i^F \sin \mathbf{y} + \boldsymbol{c}_{i-1} \sin \boldsymbol{m}_{i-1}, \alpha_i^F \cos \mathbf{y} + \boldsymbol{c}_{i-1} \cos \boldsymbol{m}_{i-1})$

        $\boldsymbol{c}_i \leftarrow ||[\alpha_i^F \sin \mathbf{y} + \boldsymbol{c}_{i-1} \sin \boldsymbol{m}_{i-1}, \alpha_i^F \cos \mathbf{y} + \boldsymbol{c}_{i-1} \cos \boldsymbol{m}_{i-1}]^T||_2$

        \# do Bayesian update for atom types

        $\alpha_i^A \leftarrow \beta_1 \left(\frac{2i-1}{n^2}\right)$

        $\mathbf{y}^A \sim \mathcal{N}\left(\alpha_i^A \left(K\mathbf{e_k} - \mathbf{1}\right), \alpha_i^A K\boldsymbol{I}\right)$

        $\boldsymbol{\theta}' \leftarrow e^{\mathbf{y}^A}\boldsymbol{\theta}_{i-1}$

        $\boldsymbol{\theta}_i \leftarrow \frac{\boldsymbol{\theta}'}{\sum_k \boldsymbol{\theta}_k'}$

    **end if**

**end for**

$\hat{\boldsymbol{A}} \sim \hat{\Psi}_A(\boldsymbol{\theta}_{n-1}^{\mathcal{M}}, t_{n-1})$ \# sample from the final probability prediction

**Return** $\hat{\boldsymbol{A}}, \hat{\Psi}_F(\boldsymbol{\theta}_{n-1}^{\mathcal{M}}, t_{n-1}), \hat{\Psi}_L(\boldsymbol{\theta}_{n-1}^{\mathcal{M}}, t_{n-1})$

---

