# OpenReview forum: "A Periodic Bayesian Flow for Material Generation"
_ICLR.cc/2025/Conference — ICLR 2025 Spotlight_

### Official Review · Reviewer_qhoR · 2024-11-01

**Soundness:** 3
**Presentation:** 3
**Contribution:** 3
**Rating:** 8
**Confidence:** 2

**Summary:**

This paper introduces a novel generative model for creating crystal structures using a Periodic Bayesian Flow. Its unique contribution is adapting Bayesian Flow Networks (BFNs) to model periodic data on non-Euclidean spaces (specifically, the hyper-torus), which is essential for the spatial symmetry of crystals. Traditional BFNs, optimized for Euclidean data, are insufficient for the periodic nature of crystals, prompting this adaptation to account for non-monotonic entropy and periodicity. The authors introduce majorly these concepts; (a) Periodic Bayesian Flow on a hyper-torus, designed for non-Euclidean spaces to improve generative modeling accuracy for crystal data. (b) Entropy Conditioning instead of time-based conditioning, which better informs the model about the generation state due to non-additive entropy dynamics. (c) Fast Sampling Algorithm and reformulations of BFN for computational efficiency, achieving approximately 100x improvement in speed over previous diffusion-based methods. Experimental results demonstrate CrysBFN's performance advantages on tasks like ab initio crystal generation and crystal structure prediction, consistently outperforming existing methods (e.g., DiffCSP and FlowMM) in accuracy, efficiency, and property statistics across datasets such as Perov-5, Carbon-24, and MP-20. The paper establishes CrysBFN as a state-of-the-art approach in generative crystal modeling, with potential applications for other data types on periodic manifolds. The approach, validated through extensive experimentation, advances both theoretical and practical methodologies for material generation tasks.

**Strengths:**

Overall, this work is thorough, well-written, and stands out as a valuable contribution to the literature on generative material models. The paper’s strengths lie in its originality, quality, clarity, and significance. Its originality is evident in the introduction of CrysBFN—the first periodic Bayesian Flow Network designed for modeling non-Euclidean crystal data on a hyper-torus. The following points highlight the paper’s most impressive strengths.

1. The paper introduces CrysBFN, the first Bayesian Flow Network designed for periodic, non-Euclidean crystal data.
2. Authors propose a novel entropy-based conditioning that enhances modeling accuracy for periodic structures.
3. Achieves a 100x speedup in sampling efficiency compared to previous diffusion-based methods.
5. State-of-the-Art Results, outperforms leading models in crystal generation accuracy and structural validity across multiple datasets.

**Weaknesses:**

1. Although not in the scope of the paper, but the paper may discuss on how CrysBFN could generalize to other non-crystal periodic or symmetrical data types. While CrysBFN shows strong results for specific crystal datasets, the paper could benefit from discussing its potential for generalization to other periodic or non-Euclidean data types beyond crystals. Including experiments or examples of how CrysBFN could extend to other symmetrical structures, such as molecular or lattice-based materials, would add depth and show broader applicability.

2. The paper introduces entropy conditioning in place of traditional time-based conditioning, which adds complexity to the model. While the necessity of entropy conditioning is discussed, providing additional comparative analysis between the two methods across varied datasets (e.g., simpler vs. more complex structures) could clarify its practical advantages and help practitioners understand when to apply entropy conditioning.

3. Though the model achieves a high sampling efficiency, the paper does not provide a detailed analysis of the overall computational cost for training and deployment. Adding a breakdown of the computational resources required, such as training time or GPU hours, would provide a clearer picture of the model’s practicality for large-scale or industrial applications.

**Questions:**

1. Given the scope of this work I do suggest that the section covering related work may be improved. The authors should do a good survey of the past work in cystal generations particularly in the field of crystal genration with other implicit generative models for crystal representations. Some of which I was able to find by searching for representation based genrative model in citations to CDVAE (Xie et. al.) paper are: 1. https://arxiv.org/abs/2306.04510, 2. https://arxiv.org/abs/2403.10846, 3. https://arxiv.org/abs/2408.07213 (kindly read and search for more). I request the authors to kindly include papers which are in the same field to address the concerns in this paper and how your research aligns or complements with these papers, so that this work becomes complete.

2. The authors are suggested to kindly improve section 2 of their work where they have mentioned some aspects of previous related work, it would be better to include how the work is different and morevoer how does the work takes the research forward from the previous work. If the space becomes an issue (As the authors have already breached the page limitation part, then kindly include a section in complementary section) kindly add a complementary section.

**Details Of Ethics Concerns:**

No concerns.

---

> ### Author Response · Authors · 2024-11-20
> **Response to Reviewer qhoR - Part 1**
>
> We greatly appreciate the reviewer’s kind recognition of our work. We believe that we have carefully addressed your concerns and responded to your questions. The detailed responses are as follows:
>
> > **W1: generalization of CrysBFN to other symmetrical data types**
>
> We sincerely appreciate your interest in exploring the potential generalization of CrysBFN to other symmetrical data types—a direction we also regard as promising for future research. Below, we offer a detailed explanation along with some initial experimental results:
>
> - **Molecular conformer generation：** Jing et al [1] proposed generating conformations by diffusing only on the periodic torsion angles and leaving the other degrees of freedom fixed, the generation quality was proved to be superior. In this setting, the proposed Periodic Bayesain Flow of CrysBFN could be directly applied to the task  for its high sampling quality and efficiency.
>
> - **All-atom 3D protein/peptide structure generation:** the side-chain geometry can be represented using up to four torsion angles corresponding to rotatable bonds between side-chain atoms $\chi_i\in[0,2\pi)^4$[2]. Similarly, CrysBFN can be applied to model these torsion angles as well.
>
> - **Experiments on peptide generation task:** Moreover, we have already conducted initial experiments applying the proposed periodic Bayesian flow to the generating the side-chain angles of peptides.  Specifically, we align the setting with torsional flow component of PepFlow [2] and evaluate the effectiveness of periodic Bayesian flow in CrysBFN. And we reported the MAE between predicted angles and ground truth angles along with the correct rate where a prediction is considered as correct with MAE < 20°.
>
>   |          | $\chi_1$ MAE | $\chi_2$MAE | $\chi_3$MAE | $\chi_4$MAE | Correct % $\uparrow$ |
>   | -------- | ------------ | ----------- | ----------- | ----------- | -------------------- |
>   | Rosseta  | 38.31        | 43.23       | 53.61       | 71.67       | 57.03                |
>   | DiffPack | 17.92        | 26.08       | 36.20       | 67.82       | 62.58                |
>   | PepFlow  | 17.38        | 24.71       | 33.63       | 58.49       | 62.79                |
>   | CrysBFN  | **13.20**    | **19.98**   | **28.11**   | **49.56**   | **66.36**            |
>
> > **W2: additional comparative analysis of entropy-conditioning mechanism across varied datasets**
>
> Thank you for your valuable suggestions! We have conducted an ablation study experiment on Perov-5 (simpler structure with only 5 atoms) and MPTS-52 (more complex structures compared to MP-20) and tabularize the results as follows:
>
> |                          | Perov-5    |        | MP-20      |        | MPTS-52    |        |
> | ------------------------ | ---------- | ------ | ---------- | ------ | ---------- | ------ |
> |                          | Match rate | RMSE   | Match rate | RMSE   | Match rate | RMSE   |
> | w/o entropy conditioning | 51.33      | 0.0753 | 52.16      | 0.0631 | 13.41      | 0.1547 |
> | CrysBFN                  | 54.69      | 0.0636 | 64.35      | 0.0433 | 20.52      | 0.1038 |
> | % Relative Improv.       | 6.55       | 15.54  | 23.37      | 31.38  | 53.02      | 32.90  |
>
> The results imply that the effectiveness of entropy conditioning. It worth noting that on more complex structures,  such as MPTS-52, the model could benefit more from the entropy conditioning.
>
> > **W3:  detailed analysis of the overall computational cost for training and deployment**
>
> That's an excellent question, and we agree that evaluating the training and deploying efficiency of CrysBFN is a crucial aspect. Below, we provide a thorough comparison of the training time to convergence across different methods, using a single 24GB NVIDIA RTX 3090 GPU in our experiments (details are provided in Appendix C):
>
> | GPU Hour | Perov-5 | MPTS-52 | MP-20 |
> | -------- | ------- | ------- | ----- |
> | DiffCSP  | 8.59    | 92.22   | 10.42 |
> | FlowMM   | 16.36   | 106.37  | 16.49 |
> | CrysBFN  | 10.19   | 85.71   | 12.31 |
>
> As for the deployment cost, we evaluate the required GPU hour to generate 10000 MP-20 crystals with the NFEs determined by the highest performance of different approaches, and the results are listed as follows:
>
> |  | CDVAE | DiffCSP | FlowMM | CrysBFN |
> | --------------------- | ----- | ------- | ------ | ------- |
> | GPU Hour/10k crystals | 8.19  | 1.55    | 3.09   | **1.09** |
>
>  We observe that: 1) FlowMM requires significantly more training and inferencing time compared to DiffCSP and CrysBFN, primarily due to its larger parameter count; 2) CrysBFN shows comparable training time with DiffCSP on simpler datasets and enjoys better training efficiency on more complex dataset, such as MPTS-52; 3) CrysBFN requires fewer GPU hours than the current method to generate 10,000 MP-20 crystals, making it more efficient for deployment.

---

> ### Author Response · Authors · 2024-11-20
> **Response to Reviewer qhoR - Part 2**
>
> > **Q1: improve the related work section and conduct a survey of the past work in crystal generations**
>
> Thank you for your suggestion! We have made an improvement on the related works section and further do a survey of the past works especially those based on implicit representations in Appendix F (due to the page limitation) as you suggested, including the papers you mentioned, and retrieved more related works. We also include the discussion on how CrysBFN complements current works by proposing a novel periodic Bayesian flow with higher sampling quality and efficiency which can be utilized in various real-world applications as mentioned in W1. We kindly hope that they could bring better clarity for you!
>
> > **Q2: more discussion on related works and how this work is different and takes the research forward**
>
> We thank the reviewer for the interest in our work! As addressed in Q1, we have improved the related work section and included a more detailed discussion in Appendix F. Additionally, we have incorporated a comparison in the preliminary section, explaining how BFNs fundamentally differ from flow matching and diffusion models, with precise guidance for each generation transition.
>
> Our paper advances research in the field from both theoretical and practical perspectives. First, we first address the challenge of extending BFN  to a non-Euclidean manifold, which sets the foundation for future exploration of BFN's potential on general manifolds. Second, our proposed frameworks serve as an efficient generative model for the real-world task of crystal generation, achieving state-of-the-art results with notable improvements in sampling speed.
>
> As you suggested, we have also elaborated on how CrysBFN contributes to addressing the periodic variable generation problem by effectively overcoming the challenges associated with constructing a periodic Bayesian flow in related work section and Appendix F. We hope these additions could provide better insight  for you!
>
> **References**:
>
> [1] Bowen Jing, Gabriele Corso, Jeffrey Chang, Regina Barzilay, and Tommi Jaakkola. Torsional diffusion for molecular conformer generation. Advances in Neural Information Processing Systems, 35:24240–24253, 2022.
>
> [2] Li, J., Cheng, C., Wu, Z., Guo, R., Luo, S., Ren, Z., ... & Ma, J. Full-Atom Peptide Design based on Multi-modal Flow Matching. In Forty-first International Conference on Machine Learning.

---

> > ### Comment · Reviewer_qhoR · 2024-11-23
> >
> > Thank you for your detailed reply. I acknowledge that the changes made by the authors address my doubts regarding the paper. I would like to maintain the current scores assigned to the paper.

---

> > > ### Author Response · Authors · 2024-11-23
> > > **Thank you for your support!**
> > >
> > > Thank you so much and we sincerely appreciate your support!

---

### Official Review · Reviewer_Yjva · 2024-11-04

**Soundness:** 2
**Presentation:** 2
**Contribution:** 3
**Rating:** 8
**Confidence:** 2

**Summary:**

This paper extends BFNs to von-Mises distributions on the product of 1-spheres (flat torus) and applies this method to modeling conditional material generation (conditioned on atom type, generate positions).

**Strengths:**

- Method seems novel and using a different model for geodesic generative modeling seems interesting.
- Results seem good and competitive.

**Weaknesses:**

- My main concern is I found the definitions and exposition on Bayesian Flow Networks very unclear.
- Some minor concerns about the experiments.

**Questions:**

About BFN extension:
  - Can the torus example in Figure 3 be explained through geodesic interpolations of $\theta_{i-1}$ and $y$?
  - I don't have sufficient background in BFN to understand the preliminaries, as it immediately uses terminology from BFNs. Can you summarize the basic BFN framework and explain why extending to periodic is not easy in 2-3 sentences in layman terms?
  - There may be missing information about the sampling in Eq (3). How is y_1 defined? How do you sample the y_i sequence? Is that based on Phi?

Experiments:
  - Why not also report stability rate for Table 2? As I understand, this is an important metric for material generation.
  - FlowMM was also proposed for efficient sampling and has a similar (but weaker) plot of Figure 4 (Match rate vs NFE). Can you add FlowMM into Figure 4?

---

> ### Author Response · Authors · 2024-11-20
> **Response to Reviewer Yjva - Part 1**
>
> We sincerely thank the reviewer for the recognition of our work and your concerns are addressed as follows:
>
> > **W1: About the definitions and exposition on Bayesian Flow Networks**
>
> Sorry for causing confusion. We have refined the presentation of BFN in the preliminary section of the revised paper version and added two toy examples (Gaussian BFN and von Mises BFN) in the anonymous code link located in the toy_examples folder. We hope they could bring greater clarity and a better understanding of how BFN works.
>
> > **W2&Q4: about the stability rate for CSP task in Table 2**
>
> Thanks for pointing it out! We calculate the stability rate for the CSP task using the same method in Table 5 following Gruver et al[1]. The results are as follows:
>
> |         | Perov-5     |          | MP-20       |          | MPTS-52     |          |
> | ------- | ----------- | -------- | ----------- | -------- | ----------- | -------- |
> |         | Meta-Stable | Stable   | Meta-Stable | Stable   | Meta-Stable | Stable   |
> | DiffCSP | 4.96        | 1.48     | 27.94       | 7.42     | 11.21       | 2.91     |
> | FlowMM  | 4.62        | 1.32     | 23.94       | 6.98     | 9.19        | 2.83     |
> | CrysBFN | **5.31**    | **1.66** | **32.41**   | **9.73** | **14.38**   | **4.08** |
>
> It could be found CrysBFN achieves consistently better performance than DiffCSP and FlowMM using stability as the CSP task metric.
>
> We also observe that the stability rates on CSP task are generally lower than ab initio generation task. We consider that this is because the CSP task has a stricter reference to compute the stability; specifically, each composition in the CSP task has a well-optimized structure based on the energy hull from DFT calculations as its reference.
>
> In contrast, ab initio-generated crystals tend to have novel compositions that are not included in the known stable structures resulting in a possibly higher reference hull energy. This is because the calculation of hull energy for a novel composition requires decomposition (like interpolation) from known stable crystals and such calculated energy may not be as low as the well-optimized reference structures.
>
> > **W2&Q5: add FlowMM in the sampling efficiency experiment**
>
> Thank you for your suggestion! We initially did not include FlowMM in Figure 4 due to its larger parameter size,  which makes a comparison based on NFE less fair. To address the reviewer's concern, we make a comparison in Figure 7 (revised version) and observe that FlowMM performs poorly in extremely low NFE settings (e.g., 20 steps), achieving only a 16.18% match rate.  In contrast, CrysBFN achieves a significantly higher match rate of 60.02% with just half that NFE (10 steps) and consistently delivers the best sampling quality across all NFEs.

---

> ### Author Response · Authors · 2024-11-20
> **Response to Reviewer Yjva - Part 2**
>
> > **Q1: Can the torus example in Figure 3 be explained through geodesic interpolations of $\theta_{i-1}$ and $y$?**
>
> Yes, you are correct! And we believe that your intuition provides an insightful perspective on explaining how the Bayesian update works on the torus. Specifically, using the notation from the paper $\theta_{i}=\\{m_{i},c_{i}\\}$, $m_{i}$ lies on the geodesics between $m_{i-1}$ and $y_i$, the location of which is determined by interpolating $m_{i-1}$ and  $y_{i-1}$ based on previous belief entropy $c_{i-1}$ and $\alpha_i$ which is the accuracy of $y_{i-1}$.
>
> > **Q2: About the preliminaries, summarizing the basic BFN framework, and explaining why extending to periodic is not easy in 2-3 sentences in layman terms?**
>
> **Summary of the basic BFN framework:** Unlike well-established SDE-based approaches, *e.g.*, Diffusion Models, and ODE-based approaches, *e.g.*, Flow Matching, Bayesian Flow Networks define a generative process driven by successive **Bayesian updates**. Based on noised samples, the uninformative prior distribution (pure noise) $\theta_{0}$ is progressively updated to new posteriors $\theta_i$ with higher confidence and more information on the data (similar to the reverse process of Diffusion models).
>
> **The challenge of extending BFN to periodic:** The uncertainties/entropies after consecutive Bayesian updates were guaranteed to be additive and monotone proved by Graves et al [2] in Euclidean space, which provides numerous desirable mathematical properties for theoretical derivation, training, and sampling. However, we find and prove that for random variables with periodicity, such additivity and monotonicity do not hold. Consequently, we need to **rethink the whole BFN framework**, construct the periodic Bayesian flow from scratch, and propose measures to tackle problems incurred in every component (including the entropy conditioning mechanism, reformulations of Bayesian flow, the fast sampling algorithm, and a numerical method for determining the accuracy schedule, etc).
>
> > **Q3: about the sampling in Eq (3) including the definition of $y_1$, the way to sample the $y_i$ sequence, and whether that is based on Phi**
>
> Sorry for causing confusion.
>
> **Definition of $y_i$ (including $y_1$):** Actually, $y_i$ is the noisy version of ground truth $x$ sampled from so-called sender distributions $y_i \sim p_S(y_i|x;\alpha_i) $ (similar to the forward process in diffusion models), where the parameters $\alpha_i$ control the signal-noise-ratio. For example, when $\alpha = 0$, $y_i$ is pure uninformative noise samples and $y_i$ is becoming more and more informative about $x$ as $\alpha$ increases. Notably, instantiations of $p_S$ for different data modalities (continuous, discrete, and periodic) are different: For continuous variables, $p_S(y|x; \alpha) = \mathcal{N}(y|x,\alpha^{-1})$. For periodic variables proposed in this paper, $p_S(y|x;\alpha)=vM(y|x,\alpha)$.
>
> **How to sample $y_i$ sequence:** As mentioned above, every $y_i$ is i**ndependently** sampled from sender distribution $\textbf{y}_i \sim p_S(\textbf{y}_i|\textbf{x};\alpha_i)$ with corresponding accuracy parameter $\alpha_i$ and instantiation according to the modality, *e.g.* Gaussian for continuous variable.
>
> **Is that based on Phi:**  As stated above, the process of sampling $y_i$ does not involve the neural network. $\alpha_i, i\in[1,n]$ are predefined as hyper-parameters which is similar to the forward process of diffusion models.
>
> **Reference**:
>
> [1] Gruver, N., Sriram, A., Madotto, A., Wilson, A. G., Zitnick, C. L., & Ulissi, Z. W. Fine-Tuned Language Models Generate Stable Inorganic Materials as Text. In The Twelfth International Conference on Learning Representations.
>
> [2] Graves, A., Srivastava, R. K., Atkinson, T., & Gomez, F. (2023). Bayesian flow networks. arXiv preprint arXiv:2308.07037.

---

> ### Author Response · Authors · 2024-11-23
> **Looking Forward to Your Feedback**
>
> Dear Reviewer Yjva,
>
> Thanks again for your insightful and thoughtful comments!
>
> As the reviewer-author discussion period is closing soon (November 26 at 11:59 pm AoE), we would like to gently remind you that we are eagerly awaiting your feedback on our response.
>
> We have improved the presentation of BFN, added a comparison to FlowMM, and reported the stability results of the crystal structure prediction task.
>
> If you have any further questions about our work, please do not hesitate to let us know.
>
> Thank you once again and we genuinely look forward to hearing from you!
>
> Best regards,
>
> Authors

---

> > ### Comment · Reviewer_Yjva · 2024-11-28
> >
> > Thank you for providing stability rates and the detailed response. With the improved clarifications on BFN and the code release, I am happy to maintain my rating.

---

> > > ### Author Response · Authors · 2024-11-28
> > > **Thank you for your support!**
> > >
> > > Thank you for your feedback and we deeply appreciate your support!

---

### Official Review · Reviewer_VEwm · 2024-11-06

**Soundness:** 2
**Presentation:** 2
**Contribution:** 2
**Rating:** 6
**Confidence:** 3

**Summary:**

This paper proposes a new crystal generative model for materials using Bayesian Flow Networks, a diffusion-like generative model but supports more types of prior for noise distributions. To effectively enforce E(3) equivariance in generation, the fractional coordinates are generated on the hyper-torus manifold defined in Jing et al. 2022. Instead of standard Gaussian noise used for generation of torsion angles, von Mises distribution is used with derived Bayesian update. To ensure the receiver belief entropy is linearly decreasing, a numerical binary search is done for determining the schedule of the sender’s accuracy (noise) level. In experiments, this method performs better than SOTA methods such as DiffCSP, FlowMM and CDVAE. When compared with diffusion-based method DiffCSP, this method excels at sampling with fewer steps (NFEs).

**Strengths:**

The major strength of the method is using Bayesian Flow Networks for crystal material generation. The von Mises distribution is used and its Bayesian update is used for training the BFN model, where previous BFN models only use Gaussian distribution. And for this special distribution, the authors identified that it is important to condition the model with the entropy of the receiver’s belief instead of the time.

The other strength is the improved sampling efficiency of BFN model as compared to diffusion-based method such as DiffCSP. But it is unclear how it compares with ODE sampling method such as flow matching or diffusion with ODE sampler.

**Weaknesses:**

Major weakness is on experiment evaluation.

- For baselines to compare, the latest DiffCSP++[1] (which gives better performanc) and MatterGen[2] are not included.
- For metrics, what about uniqueness, novelty, and stability? The goal of material generation is generating novel unique materials that are stable. Hence, these metrics are the most important to measure for use case of material generation.
- For comparing methods with same number of network forward evaluations, the sampling stepsizes should be adjusted, i.e. $\Delta t = 0.01$ if sampling $100$ steps. It’s not clear from the current description. Also diffusion models with stochastic samplers are known to need more steps in sampling. For better comparison, maybe consider flow matching (such as FlowMM) or diffusion model with ODE sampler, and with adjusted sampling stepsizes.

The presentation of BFN can be improved. Maybe with a small toy example to explain how it works. Also for introducing the parameters of von Mises distribution, might be good to refer to Figure 5 and illustrate how each parameter affect the distribution.

**Questions:**

- SE(3) or E(3) equivariance? Is reflection included when generating the fractional coordinates on the hypertorus. Only translation and rotation are mentioned to be preserved in the paper (also in Jing et al. 2022 as well), which corresponds to SE(3).
- Comparison with baselines with same number of NFEs: how are the methods evaluated, are the stepsizes $\Delta t$ adjusted by the number of sampling steps as well?

    Also for small NFEs, it makes more sense to compare with flow matching or diffusion model with ODE sampling, such as FlowMM? Since diffusion models use stochastic sampler which in general requires more steps as seen in the EDM paper[3]. How does FlowMM (which uses ODE sampler) perform with fewer steps?

- Compare with SOTA: DiffCSP++ [1] should be considered as another baseline to compare to, and also with fewer steps of sampling. Its performance seems to be better than DiffCSP. Also MatterGen-MP should be considered as a baseline for reference.
- Evaluation on metrics such as uniqueness, novelty and stability?
- Training time comparison: How does training time compare for different methods? Does the introduced von Mises distribution Bayesian update incur additional training cost per batch?

[1] Jiao, Rui, Wenbing Huang, Yu Liu, Deli Zhao, and Yang Liu. "Space group constrained crystal generation." *arXiv preprint arXiv:2402.03992* (2024).

[2] Zeni, Claudio, Robert Pinsler, Daniel Zügner, Andrew Fowler, Matthew Horton, Xiang Fu, Sasha Shysheya et al. "Mattergen: a generative model for inorganic materials design." *arXiv preprint arXiv:2312.03687* (2023).

[3] Karras, Tero, Miika Aittala, Timo Aila, and Samuli Laine. "Elucidating the design space of diffusion-based generative models." *Advances in neural information processing systems* 35 (2022): 26565-26577.

---

> ### Author Response · Authors · 2024-11-20
> **Response to Reviewer VEwm - Part 1**
>
> We thank the reviewer for the detailed review comments and insightful suggestions. We humbly believe that we have addressed the reviewer's main concerns. Consequently, we kindly hope that the reviewer could increase the rating score of our paper. The responses are as follows:
>
> > **W1&Q3: comparison with DiffCSP++ and MatterGen-MP**
>
> Thanks for pointing out two related works! However, DiffCSP++ differs from CrysBFN in the evaluation pipeline, so we conducted an additional experiment to demonstrate the effectiveness of CrysBFN within this pipeline. Regarding MatterGen-MP, there are currently no available resources to facilitate a fair comparison between MatterGen and other methods. The details are given as follows:
>
> **MatterGen-MP**: After carefully reviewing the available resources and communicating with the authors of MatterGen[1], we conclude that there is no open-sourced code or data to conduct fair comparisons between MatterGen and any method currently. The private Alex-MP-20 dataset used by MatterGen is curated from the MP and Alexandria databases, containing 607,684 DFT recalculated structures, which differs significantly from the public MP-20 dataset with only 45,231 structures. Additionally, the generated samples from MatterGen are refined using DFT, but the detailed settings for this refinement are not disclosed in their draft, making it impossible to align evaluation criteria without access to their code. Furthermore, the primary evaluation metrics of MatterGen, such as S.U.N., require a reference dataset for calculation. Without access to their private training data, reproducing or comparing these results becomes infeasible. These factors collectively limit the possibility of a direct comparison at this moment.
>
> **DiffCSP++:** It should be noted that the CSP results reported for DiffCSP++ were obtained using a different setting compared to ours. Specifically, DiffCSP++ utilizes CSPML [2] to retrieve strong initial structures with high quality, which are then refined using diffusion, whereas CrysBFN generates structures entirely from scratch. To ensure a fair comparison for CSP tasks, we implemented a retrieval-refinement version of CrysBFN (CrysBFN+CSPML) based on CSPML, similar to DiffCSP++. We provide comparisons with DiffCSP++ for CSP task in the following table:
>
> |                      | Perov-5   |            | MP-20     |            | MPTS-52   |            |
> | -------------------- | --------- | ---------- | --------- | ---------- | --------- | ---------- |
> |                      | MR        | RMSE       | MR        | RMSE       | MR        | RMSE       |
> | CSPML                | 51.84     | 0.1066     | 70.51     | 0.0338     | 36.98     | 0.0664     |
> | DiffCSP++ (w/ CSPML) | 52.17     | 0.0841     | 70.58     | 0.0272     | 37.17     | 0.0676     |
> | CrysBFN(w/ CSPML)    | **55.01** | **0.0697** | **71.03** | **0.0221** | **38.59** | **0.0597** |
>
> The results demonstrate that CrysBFN consistently outperforms DiffCSP++ in this setting. Besides, the key component introduced in DiffCSP++, *i.e.*, the space group constraint, is orthogonal to our approach and could naturally be integrated into CrysBFN to potentially enhance its performance further.
>
> > **W2&Q4:** **evaluation of uniqueness, novelty, and stability**
>
> Thank you for your suggestion and we agree that such evaluations should be included. We compute the uniqueness and novelty following FlowMM's settings while using the method in Gruver, et al[3] to calculate the stability because FlowMM does not provide its DFT code. The results are listed as follows:
>
> |         | Unique | Novelty | Stability | S.U.N |
> | ------- | ------ | ------- | --------- | ----- |
> | DiffCSP | 96.11   | 90.95    | 12.16      | 9.44   |
> | FlowMM  | 95.79   | 91.63    | 9.23       | 8.31   |
> | CrysBFN | 95.29   | 92.37    | 15.82      | 12.16  |
>
> It could be found that CrysBFN could outperform FlowMM and DiffCSP on stability and also on the metric S.U.N. which considers the uniqueness, novelty, and stability jointly.
>
> > **W3&Q2: how methods are evaluated in NFE experiment**
>
> Sorry for causing confusion. In our experiments, the step sizes $\Delta t$ are adjusted for all models by varying the global number of steps as you mentioned, i.e. $\Delta t \times N_{\text{steps}} = 1$. Specifically, keeping all hyperparameters unchanged except for the number of steps, $N_{\text{steps}}$, we conduct the training process from random model parameters, utilize saved checkpoints to sample the complete trajectory from $t = 1$ to $t = N_{\text{steps}}$, and evaluate the generated crystals for each value of $N_{\text{steps}}$.

---

> ### Author Response · Authors · 2024-11-20
> **Response to Reviewer VEwm - Part 2**
>
> > **W3&Q3: sampling efficiency comparison with FlowMM or ODE sampling of DiffCSP**
>
> We initially did not include FlowMM in Figure 4 due to its larger parameter size,  which makes a comparison based on NFE less fair.  To make our evaluation more comprehensive, we additionally include FlowMM  due to its superior performance compared to DiffCSP in Figure 7 (revised version) as you suggested. It could be observed that FlowMM performs poorly in extremely low NFE settings (e.g., 20 steps), achieving only a 16.18% match rate.  In contrast, CrysBFN achieves a significantly higher match rate of 60.02% with just half that NFE (10 steps) and consistently delivers the best sampling quality across all NFEs. This highlights that entropy-based samplers used in BFN can achieve high sampling efficiency and even outperform ODE-based samplers.
>
> > **W4: about the presentation of BFN, a small toy example, and introducing von Mises distribution parameters**
>
>  Thank you for your suggestion!
>
> **About the presentation of BFN:** As suggested, we have refined the presentation of BFN in the revised version and hope it brings greater clarity and a better understanding.
>
>  **A small toy example:** We have added two .ipynb notebooks to illustrate how Gaussian and von Mises based BFN work with minimal code under the toy_examples folder in our anonymous code repository. We kindly hope they could offer helpful insights.
>
>  **Introduction to the parameters of von Mises distribution:** We have added the explanation of how $m$ and $c$ influence the distribution shape in Appendix A (revised version) and updated Figure 5 with different mean direction parameters $m$.
>
> > **Q1: SE(3) or E(3) equivariance**
>
> Thanks a lot for pointing it out! Note that the backbones of CrysBFN directly followed previous works of DiffCSP[1] for fair comparison, which uses an EGNN-like CSPNet and it is E(3) equivariant. And this could make the learned density function is E(3) invariance for CrysBFN which includes the invariance of reflections as the reviewer mentioned. We also believe that it could be better to consider the SE(3) invariance for distinguishing the chirality in the crystals, which could be done by simply changing the backbone architecture of CrysBFN to  SE(3) equivariance. We have included the discussions as you suggested to help the community better clarify the concepts.
>
> > **Q5: training efficiency comparison**
>
> That's a great question and we agree that evaluating the training efficiency of generative models is an important aspect. Below, we provide a comprehensive comparison of the number of trained batches per GPU hour and  the training time until convergence for different methods, utilizing a single 24GB NVIDIA RTX 3090 GPU in our experimental results (details in Appendix C):
>
> | N batches / GPU Hour | Perov-5 | MPTS-52 | MP-20 |
> | -------------------- | ------- | ------- | ----- |
> | DiffCSP              | 19.2k   | 9.5k    | 15.8k |
> | FlowMM               | 13.1k   | 5.2k    | 10.2k |
> | CrysBFN              | 16.4k   | 8.1k    | 13.8k |
>
>
> | GPU Hour        | Perov-5 | MPTS-52 | MP-20 |
> | --------------- | ------- | ------- | ----- |
> | DiffCSP(12.3 M) | 8.59    | 92.22   | 10.42 |
> | FlowMM (28.3 M) | 16.36   | 106.37  | 16.49 |
> | CrysBFN(12.3 M) | 10.19   | 85.71   | 12.31 |
>
> We observe that: 1) FlowMM requires significantly more training time compared to DiffCSP and CrysBFN, primarily due to its larger parameter count; 2) CrysBFN shows comparable training time with DiffCSP on simpler datasets and enjoys better training efficiency on more complex dataset, such as MPTS-52. According to the above statistics, the von Mises Bayesian update incurs almost no additional training cost. This is attributed to the proposed fast sampling algorithm, which eliminates the need for auto-regressive Bayesian updates and instead enables parallelized tensor operations.
>
>
> References:
>
> [1] Jiao, R., Huang, W., Lin, P., Han, J., Chen, P., Lu, Y., & Liu, Y. (2024). Crystal structure prediction by joint equivariant diffusion. Advances in Neural Information Processing Systems, 36.
>
> [2] Kusaba, M., Liu, C., & Yoshida, R. (2022). Crystal structure prediction with machine learning-based element substitution. *Computational* *Materials Science*, *211*, 111496.
>
> [3] Gruver, N., Sriram, A., Madotto, A., Wilson, A. G., Zitnick, C. L., & Ulissi, Z. W. Fine-Tuned Language Models Generate Stable Inorganic Materials as Text. In *The Twelfth* *International Conference on Learning Representations*.

---

> ### Author Response · Authors · 2024-11-23
> **Looking Forward to Your Feedback**
>
> Dear Reviewer VEwm,
>
> Thanks again for your insightful and thoughtful comments!
>
> As the reviewer-author discussion period is closing soon (November 26 at 11:59 pm AoE),
> we would like to gently remind you that we are eagerly awaiting your feedback on our response.
>
> We have added a comparison with DiffCSP++ and MatterGen-MP, an evaluation on uniqueness, novelty, and stability, and clarification of NFE Experiments.
>
> If you have any further questions about our work, please do not hesitate to let us know.
>
> Thank you once again and we genuinely look forward to hearing from you!
>
> Best regards,
>
> Authors

---

> ### Author Response · Authors · 2024-11-27
> **Gentle Reminder — We Are Eagerly Awaiting Your Feedback!**
>
> Dear Reviewer VEwm,
>
> Thank you once again for your valuable comments and insights!
>
> As the reviewer-author discussion period is nearing its end (December 2nd), we wanted to kindly remind you that **we are eagerly awaiting your feedback** on our response to your comments.
>
> In our response, we have provided a detailed rebuttal towards the concerns you raised, including the addition of comparison with related works, uniqueness, novelty, and stability metrics.
>
> If you have any further questions or require clarification on any aspect of our work, please do not hesitate to let us know.
>
> Thank you for your time and effort, and we truly look forward to hearing from you!
>
> Best regards,
>
> Authors

---

> ### Comment · Reviewer_VEwm · 2024-11-27
>
> Thanks very much for your response. Most of my concerns are addressed and I have updated my score.

---

> > ### Author Response · Authors · 2024-11-28
> > **Thanks a lot !**
> >
> > Thank you for updating the score! If there is any other needed information, please don’t hesitate to let us know and we'll do our best to provide satisfactory responses.

---

### Author Response · Authors · 2024-11-25
**General Response**

We sincerely thank all reviewers for their valuable comments and suggestions. In response, we have carefully revised our paper, incorporating changes highlighted in blue text to address the concerns raised. We summarize and highlight the additional experiments and discussions conducted during the rebuttal period:

- **Peptide side-chain angles experimental results** to prove the effectiveness of the proposed periodic Bayesian flow on other symmetrical structures. (To Reviewer qhoR)
- **A comparison to DiffCSP++ within its evaluation pipeline.** (To Reviewer VEwm)
- **Sampling efficiency comparison to FlowMM** in Figure 7. (To Reviewer VEwm and Reviewer Yjva)
- **Uniqueness, novelty, stability, and S.U.N results** are added in Table 6. (To Reviewer VEwm)
- **Training efficiency comparison** across methods based on GPU hour in Table 4. (To Reviewer qhoR and Reviewer VEwm)
- Stability results of crystal structure prediction task in Table 7. (To Reviewer Yjva)
- More extensive ablation studies of entropy condition mechanism across data sets in Table 8. (To Reviewer qhoR)
- Improved presentation of Bayesian Flow Networks in Section 3, and added two .ipynb toy examples illustrating how BFN works with minimal code under toy_examples folder in our anonymous code link (To Reviewer VEwm and Reviewer Yjva).
- Revised related works section and more comprehensive discussions in Appendix F. (To Reviewer qhoR)

We kindly hope these added components effectively address the reviewers' concerns and further improve the overall quality of our paper.

---

### Meta-Review · Area_Chair_H12g · 2024-12-22

**Metareview:**

Summary
========
The paper introduces CrysBFN, a novel crystal generative model that adapts Bayesian Flow Networks (BFNs) to work with periodic data on non-Euclidean spaces, focusing on the hyper-torus manifold. The key innovations are: a) using von Mises distributions for generating torsion angles, b) implementing entropy-based conditioning, and c) developing an efficient sampling algorithm. The results (especially after rebuttal) are sufficiently extensive, demonstrating the desired model performance over rivals.

Strengths
=======
* Technical novelty in adapting BFNs for periodic, non-Euclidean spaces, which is crucial for crystal generation
* Novel use of von Mises distribution with derived Bayesian updates, advancing over standard Gaussian method
* Strong empirical results across multiple datasets

Weaknesses (mostly prior to rebuttal)
==========
* Lack of experimental evaluation against recent strong baselines (DiffCSP++, MatterGen)
* Lack of metrics for material generation (uniqueness, novelty, stability)
* Unclear exposition of BFN definitions and methodology
* Limited discussion of generalizability to other periodic or symmetrical data types

**Additional Comments On Reviewer Discussion:**

The weaknesses were addressable and have been addressed during the rebuttal period, especially those raised by reviewer VEwm, which led to increase of score.

---

### Decision · Program_Chairs · 2025-01-22

Accept (Spotlight)